# METROPOLIS-HASTINGS DISCRETE DIFFUSION: REWARD-GUIDED SAMPLING BY EXPLORING THE CLEAN DATA MANIFOLD

## ABSTRACT

Discrete diffusion models have recently emerged as a powerful class of generative models for discrete data, showing effectiveness across diverse scientific domains, such as chemistry and biology. In these fields, the notion of data quality is often well defined, for example drug-likeness in molecules, which makes reward-based guidance at inference time crucial. While reward guidance has been extensively studied for continuous diffusion models, existing approaches are either inapplicable to discrete diffusion due to their reliance on reward gradients, or ineffective because they lack local search. Some methods based on intermediate rewards are applicable to discrete diffusion but tend to underperform, since intermediate rewards are noisy due to the non-smooth nature of reward functions used in scientific domains. To address this, we propose Metropolis-Hastings Discrete Diffusion (MHDD), a method that performs effective test-time reward-guided sampling for discrete diffusion models, enabling local search without relying on intermediate rewards. The key idea is to construct a Markov chain of clean samples with the target distribution as its stationary distribution. We achieve this using the Metropolis–Hastings algorithm. However, directly applying it to discrete diffusion is infeasible due to the intractable acceptance probability. To address this, we design the proposal distribution by sequentially applying the forward and backward processes, which makes the acceptance probability tractable. Experiments on molecule and biological sequence generation with four different reward functions demonstrate that our method consistently outperforms prior approaches that rely on intermediate rewards.

## 1 INTRODUCTION

Discrete diffusion models have recently emerged as a powerful generative framework for discrete data, showing particular promise in chemistry and biology for generating complex structures such as molecules and DNA sequences. Unlike autoregressive models that assume a canonical left-to-right ordering on the data, discrete diffusion models are more naturally suited for the scientific domains such as molecule or DNA sequences which lack a natural fixed ordering. For instance, the widely used SMILES (Weininger, 1988) representation for molecules is based on heuristic rules such as depth-first search which does not use a unified fixed ordering (Lee et al., 2025).

In many applications in chemistry and biology, there are well-defined notions of *quality* for the data; for example, drug-likeness (Bickerton et al., 2012) for molecular structures or enhancer activity (Taskiran et al., 2024; Wang et al., 2025) for DNA sequences. Thus, generative models must not only produce *natural* samples that resemble the training data but also achieve high-quality scores according to such domain-specific criteria.

In the emerging regime of test-time scaling, quality considerations are incorporated by defining a reward function and optimizing it during inference through reward-guided sampling. The simplest strategy is the Best-of-N approach, where $N$ samples are generated and the one with the highest reward is selected. However, this brute-force method is inefficient since it does not perform any structured search. Recent work (Kim et al., 2025b; Wu et al., 2023; Yu et al., 2023; Bansal et al., 2023; Chung et al., 2023) has instead proposed leveraging *process rewards* or *intermediate rewards*, which are computable at intermediate steps of generation. For diffusion and flow matching models,

| | Uniform | Masked | Clean Reward | Trajectory Guidance |
|---|---|---|---|---|
| BoN | ✓ | ✓ | ✓ | ✗ |
| SMC | ✓ | ✓ | ✗ | ✓ |
| SVDD | ✓ | ✓ | ✗ | ✓ |
| SGDD | ✓ | ✗ | ✗ | ✓ |
| **MHDD** | ✓ | ✓ | ✓ | ✓ |

Table 1: Comparison of discrete diffusion inference-time scaling methods for reward-guided sampling. Our MHDD applies to all discrete diffusion frameworks and leverages the clean reward while guiding the sampling trajectory.

this approach is particularly appealing, since the expected value of the final output can be estimated from noisy intermediate states, allowing any reward function to be incorporated as an intermediate reward. By leveraging intermediate rewards, particle filtering methods can be applied to propose candidate denoised states and selectively retain the most promising ones, thereby exploring the data space more effectively through local search at each sampling step.

While the particle filtering approach with intermediate rewards is technically applicable to discrete diffusion models, it poses particular challenges for many types of chemistry and biology data. Reward functions in these domains are often non-smooth, meaning that small perturbations in the data can cause large changes in the score. For example, in molecular structures, modifying even a single element in the string representation can render the entire molecule invalid, collapsing the reward to zero, as shown in Fig. 1 (left). As a result, relying on intermediate rewards does not provide an effective local search strategy in these cases.

The key question that arises here is how to enable local search in reward-guided generation without relying on intermediate rewards, while still leveraging the characteristics of discrete diffusion models. To this end, we introduce **Metropolis–Hastings Discrete Diffusion (MHDD)** which performs iterative search on the *clean* data manifold using the Metropolis–Hastings algorithm which proposes a new candidate data and accepts or rejects it based on a computed acceptance probability.

There are two problems which prevent a straightforward application of Metropolis-Hastings to diffusion models. Firstly, since the clean data manifold is not directly modeled in discrete diffusion models, the proposal distribution for Metropolis–Hastings cannot be derived in a straightforward manner. Secondly, the Metropolis-Hastings algorithm cannot be directly applied to diffusion models due to its intractable acceptance probability. To address the first problem, we propose generating proposals through forward–backward combinations—applying the forward process to corrupt clean data followed by running the reverse process to obtain a new sample. Specifically, we first generate an initial clean data point, then apply the forward process followed by the reverse process for a certain number of steps to propose a nearby sample, and finally accept or reject it according to the Metropolis–Hastings acceptance probability. For the second problem, we show that the design of our proposal distribution makes the acceptance probability tractable, allowing Metropolis-Hastings to be directly adapted to diffusion models for the first time. An advantage of the Metropolis-Hastings algorithm is that it is guaranteed to converge to the target distribution. Due to this theoretical guarantee, a single run of the Metropolis-Hastings algorithm generates an entire sequence of samples, unlike conventional methods that generate only a single sample per run.

We validate MHDD on molecule and biological sequence generation across four different reward functions. MHDD achieves the highest reward in all settings, even for SMILES string generation where other methods fail due to inaccurate intermediate rewards.

## 2 RELATED WORK

In continuous diffusion models, reward-guided sampling is conventionally performed using gradient-based methods (Dhariwal & Nichol, 2021; Ho & Salimans, 2022; Chung et al., 2023; Song et al., 2023; Rozet et al., 2024; Bansal et al., 2023; Yoon et al., 2025; Kim et al., 2025b; Wu et al., 2023) which offer strong guidance towards high-reward regions. However, gradient-based approaches cannot be applied to discrete diffusion models as gradients are ill-defined in discrete spaces and it is not theoretically valid to add a continuous gradient to a discrete objective.

**Training-Free Reward-Guided Sampling for Discrete Diffusion.** Recently, inference-time scaling methods for discrete diffusion models have been proposed to tackle reward-guided sampling. A comparison of these methods and our method is shown in Tab. 1.

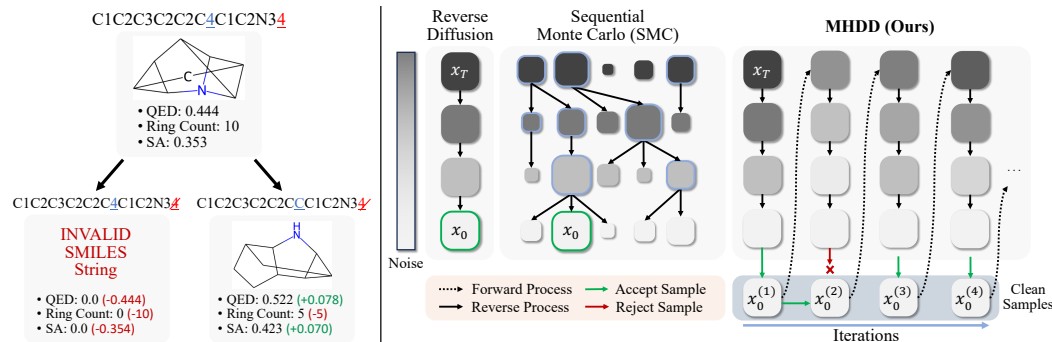

Figure 1: **Left:** In scientific applications, the rewards defined on discrete spaces are highly sensitive to small perturbations. A one-character change to a SMILE string can result in an invalid string with zero reward. Properties such as QED, ring count, and synthetic accessibility (SA) can also vary significantly even when changing only one or two tokens. **Right:** Reverse diffusion and typical inference-time scaling methods (Kim et al., 2025b; Li et al., 2024) such as SMC (Kim et al., 2025b) rely on guiding samples through noise levels by constructing a Markov chain beginning with pure noise and ending with clean samples. Our MHDD constructs a Markov chain consisting of only clean samples by successively applying the forward and reverse processes sequentially at each step. This formulation bypasses the need for intermediate rewards by evaluating the reward directly on clean samples while leveraging information from past samples for guidance.

The simplest method is Best-of-N (BoN) sampling (Stiennon et al., 2020), which generates $N$ samples independently and selects the one with the highest reward. Due to its simplicity, BoN is applicable to all types of discrete diffusion models. However, BoN does not guide the denoising trajectory using the reward, resulting in inefficient search, especially when high-reward samples lie in low-density regions unlikely to be sampled by the model.

On the other hand, particle-based methods (Ma et al., 2025; Kim et al., 2025a; Singhal et al.; Li et al., 2024), such as SMC (Doucet et al., 2001; Naesseth et al., 2019; Moral, 2004) and SVDD (Li et al., 2024), can be applied to discrete diffusion and incorporate reward signals *during* the denoising process through a reward-based resampling step. SMC (Doucet et al., 2001; Naesseth et al., 2019; Moral, 2004), take $N$ particles (samples) at each step and select a subset of the particles to keep while throwing the rest away. By only keeping high-reward particles, these methods encourage local exploration around potentially promising samples. SVDD (Li et al., 2024) exploits local search around high-reward samples by generating multiple candidate samples at each timestep and retaining only one sample by sampling the candidates with probability proportional to the reward. A key challenge with these methods is that they require computing **intermediate rewards** on noisy samples rather than on fully denoised samples. To address this, these approaches exploit the diffusion model's ability to predict an approximation of the clean sample $x_0$ given a noisy sample $x_t$, and compute the rewards on this $x_0$ prediction instead. This enables reward-guidance *during* the denoising process, but also inherently assumes that the predicted $x_0$ is a good approximation of the true clean sample since exploration will be performed locally around samples with high intermediate rewards, an assumption which does not hold in many scientific applications.

Recently, Chu et al. (2025) proposed SGDD which uses the split Gibbs sampler (Vono et al., 2019) by alternating between denoising steps and running MCMC to optimize intermediate noisy samples. However, SGDD applies *only* to uniform diffusion models where intermediate noisy samples can be treated as clean samples. The characteristics of uniform discrete diffusion is discussed in more detail in Sec. 4. Furthermore, SGDD still relies on computing the reward on the intermediate noisy samples during the MCMC optimization step.

In this work, we bypass the need for intermediate rewards by using the Metropolis-Hastings algorithm (Robert & Casella, 2009) to construct a Markov chain of *clean* samples that converges to the desired target distribution. Our method is applicable to both uniform and masked discrete diffusion models, and only requires the computation of **clean rewards** on clean samples, which can then be used to efficiently guide the Markov chain (Tab. 1).

**Training-Based Reward-Guided Sampling for Discrete Diffusion.** Unlike continuous diffusion, discrete diffusion models cannot leverage gradient signals due to the non-differentiable nature of

discrete spaces. As such, only gradient-free approaches can be applied to discrete diffusion models. Guidance methods (Nisonoff et al.; Schiff et al., 2024) have also been developed for diffusion models but require training a classifier on noisy data. Due to its non-differentiability, RLFT approaches are often used in discrete diffusion (Rector-Brooks et al.; Wang et al., 2025; Zekri & Boullé, 2025) but are difficult to train since non-differentiability forces fine-tuning to be done using policy gradients based on highly complex reward landscapes (Uehara et al., 2025). Crucially, modifying the model weights also runs the risk of deviating from the pretrained prior, affecting the naturalness of the generated samples and further complicating training.

# 3 BACKGROUND

Diffusion models (Sohl-Dickstein et al., 2015; Ho et al., 2020; Song et al., 2020) learn to reverse a forward Markov process. Although originally designed for continuous spaces, diffusion models have also been successfully applied to discrete state spaces (Austin et al., 2021; Campbell et al., 2022; Lou et al., 2024; Sahoo et al., 2024; Schiff et al., 2024; Shi et al., 2024). Discrete diffusion models follow either a discrete-time or continuous-time framework, and models within each framework can be divided into masked and uniform discrete diffusion models.

## 3.1 DISCRETE-TIME VS. CONTINUOUS-TIME DISCRETE DIFFUSION

**Discrete-Time Discrete Diffusion.** These discrete diffusion models are characterized by forward transition matrices $\boldsymbol{Q}_t$. Let $\overline{\boldsymbol{Q}}_t = \boldsymbol{Q}_1 \boldsymbol{Q}_2 \cdots \boldsymbol{Q}_t$. These transition matrices define the forward process:

$$p_{t|t-1}(x_t|x_{t-1}) = \text{Cat}\left(x_t; p = x_{t-1}\boldsymbol{Q}_t\right),$$
$$p_t(x_t|x_0) = \text{Cat}\left(x_t; p = x_0\overline{\boldsymbol{Q}}_t\right),$$

where $\text{Cat}(\cdot; p)$ denotes the categorical distribution with probabilities given by $p$.

While the reverse process probability $p(x_{t-1}|x_t)$ is not directly tractable, by additionally conditioning on $x_0$, the reverse process can be derived in closed form as

$$p(\mathbf{x}_{t-1}|x_t, \mathbf{x}_0) = \text{Cat}\left(x_{t-1}; p = \frac{x_t\boldsymbol{Q}_t^\top \odot x_0\overline{\boldsymbol{Q}}_{t-1}}{x_0\overline{\boldsymbol{Q}}_t x_t^\top}\right).$$

A neural network $p_\theta(x_0|x_t) \approx p(x_0|x_t)$ is learned and denoising is performed by the following parameterization:

$$p_\theta(x_{t-1}|x_t) \propto \sum_{x_0} q(x_{t-1}, x_t|x_0)p_\theta(x_0|x_t).$$

Notably, the learned neural network $p_\theta(x_0|x_t)$ predicts a distribution over $x_0$ given $x_t$ which enables sampling $x_0$-predictions $\hat{x}_0(x_t) \sim p_\theta(x_0|x_t)$.

**Continuous-Time Discrete Diffusion.** Campbell et al. (2022) proposed a continuous-time framework based on Continuous-Time Markov Chains (CTMC) where state transitions can occur at any time. Instead of defining transition matrices, CTMC-based discrete diffusion defines a forward transition *rate* matrix $\boldsymbol{R}_t$ which defines the infinitesimal transition probability between two timesteps:

$$p(x_t|\tilde{x}_{t-\Delta t}) = \delta_{x_t, \tilde{x}_{t-\Delta t}} + \boldsymbol{R}_t(\tilde{x}_{t-\Delta t}, x_t)\Delta t + o(\Delta t).$$

In practice, a base rate matrix $\boldsymbol{R}_b$ is chosen based on the choice of forward process, and $\boldsymbol{R}_t$ is set as $\boldsymbol{R}_t = \beta(t)\boldsymbol{R}_b$ for some user-defined scheduling function $\beta(t) \in \mathbb{R}$. This parameterization of $\boldsymbol{R}_t$ is crucial in obtaining a tractable, closed-form expression for $p_t(\cdot|x_0)$:

$$p_t(x_t = j|x_0 = i) = \left(Q \exp\left[\Lambda \int_0^t \beta(s)ds\right] Q^{-1}\right)_{ij},$$

where $\boldsymbol{R}_b = Q\Lambda Q^{-1}$ is the eigendecomposition of $\boldsymbol{R}_b$ and $\exp(\cdot)$ is the element-wise exponential.

The reverse process is performed by learning a reverse transition rate matrix $\hat{\boldsymbol{R}}_t$ and sampling from

$$p(\tilde{x}_t|x_{t+\Delta t}) = \delta_{\tilde{x}_t, x_{t+\Delta t}} + \hat{\boldsymbol{R}}_t(x_{t+\Delta t}, \tilde{x}_t)\Delta t + o(\Delta t).$$

By using a continuous-time framework, advanced sampling strategies such as predictor-corrector methods (Campbell et al., 2022; Zhao et al., 2024) and planning (Liu et al.; Peng et al., 2025) have been proposed. Score-based discrete diffusion models (Meng et al., 2022; Sun et al.; Lou et al., 2024) have also been introduced by leveraging the CTMC framework.

## 3.2 Masked vs. Uniform Discrete Diffusion

Discrete diffusion models allow the user to choose the transition matrix. The two most common choices of transition matrices result in masked diffusion models (MDMs) (Austin et al., 2021; Sahoo et al., 2024; Shi et al., 2024) and uniform state models (USMs) Austin et al. (2021); Schiff et al. (2024).

**Masked Diffusion Models (MDMs).**  MDMs define the forward process by progressively replacing tokens with a special mask token. At higher noise levels, more tokens are replaced, and the denoising model learns to recover the original sequence from these masked inputs. Notably, samples at intermediate time steps are not valid samples because of the mask tokens, which are not present in the dataset.

**Uniform State Models (USMs).**  USMs replace each token with a randomly chosen token from the vocabulary as noise increases. This creates a uniform corruption process in which every other possible token substitution is equally likely. Unlike MDMs, samples at intermediate time steps may constitute valid samples since all tokens are taken from the dataset.

## 4 Metropolis-Hastings Discrete Diffusion

Let the reward function be denoted by $r(\cdot) : \mathcal{X} \to \mathbb{R}$ where $\mathcal{X}$ is the domain, and $\Delta(\mathcal{X})$ denote the set of all probability distributions defined on $\mathcal{X}$. Given a reward function $r(x)$, our objective is to generate samples with high rewards while maintaining naturalness by leveraging a pretrained generative model $p^{\text{pre}}(\cdot)$. To accomplish this, we sample from the reward-weighted distribution $p_\beta(x)$:

$$p_\beta(x) := \underset{p \in \Delta(\mathcal{X})}{\arg\max} \mathbb{E}_{x \sim p(\cdot)} \left[ r(x) \right] - \beta D_{\text{KL}}(p(\cdot) \| p^{\text{pre}}(\cdot)) \propto \exp(r(x)/\beta) p^{\text{pre}}(x),$$

where $\beta$ is a hyperparameter controlling the "naturalness" of the samples through KL-regularization with the pretrained model.

Most previous methods such as particle-based methods perform guidance by utilizing intermediate rewards $r(\hat{x}_0(x_t))$ computed from the $x_0$-prediction $\hat{x}_0(x_t)$ at $x_t$ as an approximation to the clean sample. However, intermediate rewards are often inaccurate and noisy in many scientific applications due to the non-smooth reward functions. In scientific applications, even a slight perturbation can significantly impact the reward. For instance, when generating molecules based on the SMILES (Weininger, 1988) representation, modifying a single token on a high-reward molecule can result in an invalid molecule with zero reward or a molecule with very different properties, as shown in Fig. 1 (left). This is in stark contrast to rewards defined on continuous space where a slight perturbation smoothly affects the reward (small perturbations to some pixel values of an image leave the semantics of the image unchanged). Due to the sensitivity of the reward, the $x_0$ prediction must be perfectly accurate, otherwise the intermediate rewards will be uninformative. A sample whose intermediate reward is zero may be prematurely removed by an SMC algorithm even though changing a single token may yield a high-reward sample (Fig. 1, left).

In these cases, it is more beneficial to leverage only clean reward $r(x_0)$ computed on a clean sample $x_0$ rather than on an approximation $\hat{x}_0(x_t)$. One such method leveraging clean rewards is Best-of-N (BoN) sampling (Stiennon et al., 2020). However, BoN does not provide guidance during the denoising process, resulting in inefficient exploration. Its effectiveness is therefore limited to what the pretrained model can already generate: if high-reward samples lie in low-density regions of the model's distribution, they are unlikely to ever be produced, even when many samples are drawn. A natural question arises: **How can we leverage clean rewards while using information from past samples for guidance?**

Our proposed method, Metropolis-Hastings Discrete Diffusion (MHDD), answers this question by using the Metropolis-Hastings (MH) algorithm to construct a Markov chain of clean samples that converges to the reward-weighted distribution $p_\beta(x_0)$. By using only clean samples, MHDD leverages accurate clean rewards. By using the MH algorithm to iteratively refine the samples, the clean rewards of past samples can be used for guidance.

### 4.1 EXPLORING THE CLEAN DATA MANIFOLD

Previous methods require intermediate rewards because the denoising trajectory of diffusion models output clean samples only at the very end. A natural solution is to instead explore the clean data manifold by constructing a chain of *clean* samples $\{x_0^{(i)}\}_i$, which converges to the target distribution $p_\beta(x_0)$. At each iteration, rewards can then be computed directly on clean samples, yielding clean rewards $r(x_0)$ which can then be used to guide the chain towards the target distribution. To accomplish this, we propose to use the Metropolis-Hastings (MH) algorithm.

Suppose we want to sample from the (potentially unnormalized) target distribution $p_\beta(x_0)$. We first define a proposal distribution $q(x_0'|x_0)$ to generate the next candidate sample $x_0' \sim q(x_0'|x_0)$ given the current sample $x_0$. After generating the proposal candidate, we decide whether or not to accept this proposal with probability $A(x_0'|x_0)$ defined as

$$A(x_0'|x_0) = \min\left(1, \alpha\right), \quad \alpha = \frac{p_\beta(x_0')q(x_0|x_0')}{p_\beta(x_0)q(x_0'|x_0)}.$$

With a slight abuse of notation, we use $A(x_0'|x_0)$ interchangeably with $A$ when the choice of $x_0'$ and $x_0$ are not important. The MH algorithm can be shown to construct a Markov chain whose stationary distribution is the target distribution. For more details and proof of convergence on the MH algorithm, please refer to App. A. To generate samples from the target reward-weighted distribution $p_\beta(x_0) \propto \exp(r(x_0)/\beta)p(x_0)$, the acceptance probability is as follows:

$$A(x_0'|x_0) = \min(1, \alpha), \quad \alpha = \exp\left(\frac{r(x_0') - r(x_0)}{\beta}\right)\frac{p(x_0')q(x_0|x_0')}{p(x_0)q(x_0'|x_0)}. \tag{1}$$

The difficulty in applying the MH algorithm to diffusion models is the intractibility of the acceptance probability. Since $p(x_0)$ is intractable, the acceptance probability cannot be computed.

### 4.2 FORWARD-BACKWARD PROPOSAL DISTRIBUTION

To bypass the computation of $p(x_0)$, we define the proposal distribution in such a way that the proposal probabilities $q(x_0|x_0')$ and $q(x_0'|x_0)$ cancel out $p(x_0)$ and $p(x_0')$. We define the proposal distribution $q(x_0'|x_0)$ using a forward-backward diffusion process as follows. Starting from the current clean sample $x_0$, we choose a random time $t \sim \mathcal{U}(t_{\text{lo}}, t_{\text{hi}})$ (where $t_l$ and $t_h$ are user-defined parameters) and apply the forward process to obtain a noisy auxiliary sample $x_t \sim p_t(\cdot|x_0)$. Then, we run the reverse process to obtain a new clean sample $x_0' \sim p(\cdot|x_t)$ This proposal distribution is defined by

$$q(x_0'|x_0) := p_t(x_t|x_0)p(x_0'|x_t), \tag{2}$$

where $t \sim \mathcal{U}(t_{\text{lo}}, t_{\text{hi}})$ and $x_t \sim p(\cdot|x_0)$ are resampled at each step. Although it may appear that this Markov chain is time-inhomogeneous, it is secretly *time-homogeneous*. To see this, we simplify proposal distribution to an equivalent time-homogeneous proposal distribution. The probability of proposing $x_0'$ given that we are currently at state $x_0$ is $p_t(x_t|x_0)p(x_0'|x_t)$ with probability $\frac{1}{t_{\text{hi}} - t_{\text{lo}}}$. To obtain the probability $q(x_0'|x_0)$, we simply condition the proposal probability on $t$ and $x_t$ and integrate over all possible $t$ and $x_t$:

$$q(x_0'|x_0) = \int_{t_{\text{lo}}}^{t_{\text{hi}}} \int_{\mathbb{X}_t} p_t(x_t|x_0)p(x_0'|x_t)f(t)dx_t dt$$
$$= \mathbb{E}_{t\sim\mathcal{U}(t_{\text{lo}}, t_{\text{hi}})}\mathbb{E}_{x_t\sim p_t(\cdot|x_0)}[p(x_0'|x_t)],$$

which is time-homogeneous. Thus, we still have an equivalent time-homogeneous Markov chain and standard convergence proofs (refer to App. A) apply.

However, we find that running a one-step reverse process results in an inaccurate $x_0'$ in practice. Instead, we run an $M$-step reverse process on $x_t$ to obtain $(x_{t_M}, x_{t_{M-1}}, \ldots, x_1, x_0') \sim \prod_{i=1}^M p(x_{t_{i-1}}|x_{t_i})$ where $t = t_M > \cdots > t_0 = 0$. We then discard the $x_{t_i}$ and retain only $x_0'$. This proposal distribution can be interpreted as exploring locally around $x_0$.

**Algorithm 1 MHDD** (Metropolis-Hastings Discrete Diffusion)

1: $x_0^{(1)} \sim p(x_T) \prod_{t=1}^T p(x_{t-1}|x_t)$
2: **for** $k = 1, \ldots, K$ **do**
3: $\quad t_0 \sim \mathcal{U}(t_{\text{lo}}, t_{\text{hi}})$
4: $\quad x_{t_0}^{(k)} \sim p_{t_0}(\cdot|x_0^{(k)})$
5: $\quad \tilde{x}_0^{(k+1)} \sim \prod_{i=1}^M p(x_{t_{i-1}}^{(k)}|x_{t_i}^{(k)})$
6: $\quad \alpha \leftarrow \exp((r(\tilde{x}_0^{(k+1)}) - r(x_0^k))/\beta)$
7: $\quad A \leftarrow \min(1, \alpha)$
8: $\quad \rho \sim \mathcal{U}(0, 1)$
9: $\quad x_0^{(k+1)} \leftarrow \tilde{x}_0^{(k+1)}$ if $\rho < A$ else $x_0^{(k)}$
10: **end for**
11: **return** $\{x_0^{(K/2)}, \ldots, x_0^{(K)}\}$

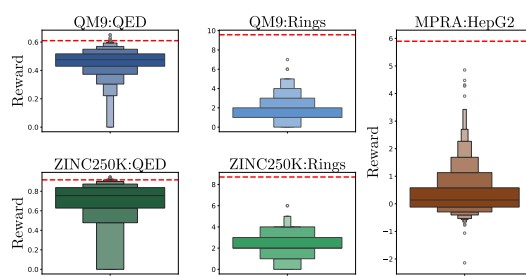

Figure 2: Reward distributions of the pretrained USM. The red dotted line represents the average reward achieved by MHDD. For ring count and HepG2, the pretrained model reward distribution has low density at higher rewards, resulting in degraded performance for BoN sampling.

Using this proposal distribution, we have

$$\alpha = \exp\left(\frac{r(x_0') - r(x_0)}{\beta}\right) \frac{p(x_0')p(x_t|x_0')p(x_0|x_t)}{p(x_0)p(x_t|x_0)p(x_0'|x_t)}$$

$$= \exp\left(\frac{r(x_0') - r(x_0)}{\beta}\right) \frac{p(x_0')p(x_t|x_0')p(x_0|x_t, x_0')}{p(x_0)p(x_t|x_0)p(x_0'|x_t, x_0)}$$

$$= \exp\left(\frac{r(x_0') - r(x_0)}{\beta}\right).$$

Thus, the acceptance probability simplifies to

$$A(x_0'|x_0) = \min(1, \alpha), \quad \alpha = \exp\left(\frac{r(x_0') - r(x_0)}{\beta}\right). \tag{3}$$

The detailed derivation of the acceptance probability can be found in App. B. Importantly, the acceptance probability can now be efficiently computed.

MHDD begins by running the full reverse process to obtain a clean initial sample $x_0^{(1)}$. At each step, we generate a candidate sample using our forward-backward diffusion process and choose to accept or reject the candidate based on the MH acceptance probability given in Eq. 3, as shown in Fig. 1 (right). This results in a chain of clean samples whose stationary distribution is the reward-weighted distribution. While MH is guaranteed to construct a Markov chain whose stationary distribution is $p_\beta(x_0)$, initial samples may not come from the stationary distribution. Thus, we discard the first half of the chain and take as many samples as desired from the latter half at equal intervals. The practice of "throwing away" the initial samples of a Markov chain is known as **burn-in** (Robert & Casella, 2009; Murphy, 2023). The full algorithm for MHDD is shown in Alg. 1.

One advantage of MHDD is once it converges to the stationary distribution, we can take arbitrarily many samples from the chain since the stationary distribution is the target reward-weighted distribution. In contrast, other inference-time scaling algorithms can only produce a limited number of samples. For instance, BoN can only produce one sample per run while SMC can only produce as many samples as the number of particles used, which is limited by the memory budget.

## 5 EXPERIMENTS

We conduct experiments on molecule generation with QM9 (Ramakrishnan et al., 2014) and ZINC250K (Irwin et al., 2012), and biological sequence design using the MPRA (Gosai et al., 2023) dataset. We pretrain four discrete diffusion models on each of the dataset:

- **MDM (Sahoo et al., 2024):** A masked discrete diffusion model using discrete-time ancestral sampling.
- **USM (Schiff et al., 2024):** A uniform state discrete diffusion model using discrete-time ancestral sampling.

- **SEDD-M (Lou et al., 2024)**: A score-based CTMC masked discrete diffusion model.
- **SEDD-U (Lou et al., 2024):** A score-based CTMC uniform discrete diffusion model.

We compare the following training-free reward-guided sampling methods for discrete diffusion:

- **Pretrained models:** Samples are generated using the pretrained model.
- **Best-of-N (BoN):** $N$ samples are generated and the one with the highest reward is selected.
- **SMC (Doucet et al., 2001):** The representative derivative-free particle-based sampling method which approximates the target distribution by updating and resampling a set of $N$ particles at each step using the intermediate reward.
- **SVDD (Li et al., 2024):** $N$ candidate samples are generated at each step and a sample is selected by randomly choosing a sample with probability proportional to its intermediate reward.
- **SGDD (Chu et al., 2025):** A posterior sampling method specifically tailored for uniform CTMC discrete diffusion models which uses split Gibbs sampling (Vono et al., 2019). Since this method is based on the forward process of uniform CTMC discrete diffusion models, we only evaluate SGDD on SEDD-U and not on the discrete-time model USM.
- **MHDD(Ours):** We run Algorithm 2 to construct a Markov chain converging to $p_\beta(x_0)$ and draw multiple samples from the chain.
- **MHDD-B (Ours):** MHDD with batching by running $B$ chains in parallel while keeping the total NFE fixed by reducing the number of iterations in each Markov chain. MHDD-B is faster than MHDDbut sometimes results in lower reward. We use a batch size of 8 for QED and SA rewards, and a batch size of 4 for ring count and HepG2.

We match the total diffusion model NFE for each method. More experiment details can be found in App. C.

**Molecule Generation.** We test our method on two molecule datasets: QM9 (Ramakrishnan et al., 2014) and ZINC250K (Irwin et al., 2012). QM9 is a dataset consisting of ∼133,000 small organic molecules, and ZINC250K is a dataset of 250,000 commercially available compounds. The molecules in both datasets are represented as SMILES strings (Weininger, 1988). For our reward functions, we use **QED** (Bickerton et al., 2012), ring count **(Rings)**, and synthetic accessibility **(SA)** (Ertl & Schuffenhauer, 2009). QED measures the drug-likeness of a compound based on eight widely used molecular properties (number of hydrogen bond donors/acceptors, molecular polar surface area, number of aromatic rings, etc.). Higher QED values indicate higher drug-likeness. Ring count measures the number of rings in the symmetrized smallest set of smallest rings (SSSR). Finally, SA measures the ease of synthesis of drug-like molecules through a combination of known common structural features in known synthesized molecules, and a penalty based on complex structural features of the molecule. SA takes on a value bewteen 1 and 10 where higher values indicate that the molecule is harder to synthesize. In this work, SA is converted to a reward function by applying the renormalization $(10 - SA)/9$ so that higher values indicate better performance. For all rewards, higher is better.

**Biological Sequence Generation.** We pretrain the discrete diffusion models on the DNA dataset provided by Gosai et al. (2023) (which we refer to as MPRA) which measures the enhancer acitivity of ∼700,000 DNA sequences using massively parallel reporter assays (MPRA). An Enformer model (Avsec et al., 2021) is trained to predict the enhancer activity level in the HepG2 cell line. This predicted HepG2 activity is used as the reward function. We use the model trained on two different subsets of data and use one exclusively to provide the guidance signal during sampling and the other for evaluation. Higher predicted HepG2 activity indicates better performance.

**Results.** The quantitative results of reward-guided generation are shown in Tab. 2. As shown, MHDDor MHDD-B achieves the best reward in all cases.

SMC and SVDD require intermediate rewards which are often inaccurate. As such, these methods sometimes only yield slightly improved results over the pretrained model and, in some cases, slightly worse results due to inaccurate exploration. The performance of these methods is also highly reliant on both the smoothness of the reward function and the diffusion model's accuracy in $x_0$ prediction. MHDD, on the other hand, only use accurate clean rewards for guidance, resulting in accurate and efficient guidance.

| | | QM9 | | | ZINC250K | | | MPRA |
| | | QED | Rings | SA | QED | Rings | SA | HepG2 |
|---|---|---|---|---|---|---|---|---|
| MDM | Pretrained | 0.461±.138 | 2.755±3.432 | 0.558±.0227 | 0.663±.323 | 2.020±2.488 | 0.742±.323 | 0.442±1.767 |
| | BoN | 0.580±.073 | 5.479±1.286 | 0.818±.129 | 0.854±.121 | 3.854±1.923 | 0.862±.121 | 1.842±2.093 |
| | SMC | 0.512±.144 | 2.803±1.713 | 0.759±.271 | 0.637±.310 | 2.128±2.594 | 0.769±.310 | 3.087±2.975 |
| | SVDD | 0.567±.117 | 2.849±1.462 | 0.836±.174 | 0.776±.255 | 2.490±1.848 | 0.754±.255 | 2.319±2.654 |
| | **MHDD** | **0.610**±.085 | **10.00**±0.933 | **0.913**±0.077 | 0.910±0.060 | **8.091**±2.328 | **0.905**±0.060 | **5.259**±0.849 |
| | **MHDD-B** | **0.610**±.083 | 8.678±2.531 | 0.911±0.095 | **0.914**±0.049 | 6.032±1.918 | 0.903±0.073 | 5.127±1.428 |
| USM | Pretrained | 0.461±.152 | 2.032±1.341 | 0.620±0.223 | 0.739±0.270 | 2.598±1.883 | 0.768±0.189 | 0.351±1.541 |
| | BoN | 0.600±.054 | 5.044±1.267 | 0.839±0.097 | 0.901±0.065 | 4.184±1.307 | 0.889±0.048 | 1.753±1.965 |
| | SMC | 0.485±.170 | 1.957±1.212 | 0.660±0.203 | 0.750±0.270 | 2.633±1.828 | 0.785±0.184 | 0.743±2.048 |
| | SVDD | 0.454±.157 | 2.135±1.460 | 0.624±0.217 | 0.750±0.260 | 2.562±1.861 | 0.773±0.195 | 0.695±1.920 |
| | **MHDD** | **0.610**±.085 | **9.570**±0.791 | **0.908**±.063 | **0.917**±0.050 | **8.703**±3.454 | **0.898**±0.060 | **5.897**±1.499 |
| | **MHDD-B** | 0.609±.081 | 9.240±2.401 | 0.915±0..088 | 0.910±0.063 | 6.310±2.494 | 0.895±0.073 | 5.292±1.876 |
| SEDD-M | Pretrained | 0.460±.143 | 2.305±3.318 | 0.588±0.240 | 0.669±0.328 | 2.350±2.648 | 0.731±0.243 | 0.373±1.631 |
| | BoN | 0.582±.069 | 5.241±2.441 | 0.831±0.123 | 0.852±0.119 | 4.057±2.022 | 0.857±0.088 | 1.874±2.243 |
| | SMC | 0.461±.162 | 2.425±3.365 | 0.599±0.263 | 0.669±0.338 | 2.251±2.856 | 0.745±0.224 | 0.386±1.639 |
| | SVDD | 0.450±.159 | 2.378±3.243 | 0.575±0.244 | 0.666±0.326 | 2.362±2.707 | 0.739±0.245 | 0.456±1.824 |
| | **MHDD** | **0.619**±.096 | **9.792**±3.402 | **0.894**±0.113 | **0.875**±0.150 | **10.026**±4.305 | **0.885**±0.088 | **7.153**±1.439 |
| | **MHDD-B** | 0.594±.098 | 8.977±3.041 | 0.821±0.262 | 0.846±0.189 | 5.677±5.179 | 0.863±0.162 | 5.991±1.643 |
| SEDD-U | Pretrained | 0.458±.154 | 1.654±2.355 | 0.637±0.230 | 0.741±0.266 | 2.524±1.753 | 0.771±0.189 | 0.455±1.755 |
| | BoN | 0.583±.066 | 3.862±1.996 | 0.843±0.112 | 0.904±0.045 | 4.056±1.241 | 0.889±0.049 | 1.729±2.216 |
| | SMC | 0.546±.113 | 2.718±2.790 | 0.811±0.196 | 0.850±0.174 | 2.610±1.684 | 0.847±0.136 | 3.334±3.337 |
| | SVDD | 0.518±.120 | 2.482±2.624 | 0.779±0.234 | 0.809±0.217 | 2.759±1.784 | 0.817±0.148 | 3.189±3.330 |
| | SGDD | 0.536±.121 | 2.644±2.791 | 0.684±0.211 | 0.844±0.152 | 2.535±1.627 | 0.847±115 | 9.240±2.050 |
| | **MHDD** | **0.619**±.080 | **7.488**±5.376 | 0.886±0.075 | **0.922**±0.073 | **5.499**±2.614 | **0.898**±0.074 | **10.09**±2.637 |
| | **MHDD-B** | 0.572±.068 | 7.213±4.325 | **0.908**±0.120 | 0.894±0.112 | 3.625±0.870 | 0.883±0.088 | 9.350±1.762 |

Table 2: Average reward (with 95% confidence intervals) for each method and pretrained discrete diffusion model. Higher is better. **Bold** indicates the best method, and underline denotes the second best. MHDD achieves the best average reward across all datasets and discrete diffusion models.

While BoN performs well on QED and SA for QM9 and ZINC250K, its performance is not as good on the HepG2 activity reward for MPRA and the ring count reward for QM9 and ZINC250K. This is due to the lack of trajectory guidance: BoN performs well only for settings where high-reward samples are likely to be sampled by the pretrained model and performs worse in cases where high-reward samples lie in low-density regions of the pretrained model. As shown in Fig. 2, for ring count and HepG2 rewards, the high-reward samples generated by MHDD (red dashed line) lie in low-density regions unlikely to be sampled from by the pretrained model. MHDD, which uses the forward-backward diffusion process to guide the samples toward high-reward regions, is able to achieve high rewards across all datasets and reward functions even when high-reward samples lie in low-density regions of the pretrained model.

Although SGDD achieves the second-best result on the MPRA dataset, it is only applicable to uniform CTCM discrete diffusion models whereas our MHDD is applicable to all discrete diffusion models. Furthermore, SGDD still relies on computing intermediate rewards, limiting its performance on datasets such as QM9 and ZINC250K where the intermediate rewards are inaccurate.

**Wall-Clock Time.** We include comparisons of wall-clock time in Tab. 3 and performance under matched wall-clock time in Tab. 4 and Tab. 5. MHDD-B (MHDD with batching) is significantly faster than MHDD, as shown in Tab. 3 .Notably, MHDD-B yields the best performance, is faster than SMC and, at equal batch sizes, roughly as fast as BoN as well. As shown in Tab. 4 and Tab. 5, MHDD significantly outperforms all other baselines in ring count and HepG2 rewards while being comparable with BoN in QED and SA. Furthermore, when we match the batch sizes, MHDD outperforms BoN in all rewards.

# 6 CONCLUSION

We propose **MHDD**, a novel training-free reward-guided sampler for discrete diffusion based on Metropolis-Hastings sampling on the clean data manifold. The key idea is to avoid reliance on noisy intermediate rewards during the generative process by instead starting from a generated sample and performing a Markov chain on the clean data manifold with a carefully designed proposal distribution. Our proposal distribution, modeled through a forward–backward combination, makes the acceptance probability tractable. Experiments on molecular and biological sequence genera-

| | MHDD | MHDD-B | BoN | BoN | SMC | SVDD |
|---|---|---|---|---|---|---|
| Time (s) | 3029 | 334 | 334 | 359 | 359 | 264 |
| Batch Size | 1 | 8 | 78 | 8 | 14 | 886 |
| NFE | 1024 | 1024 | 5750 | 1024 | 1024 | 65536 |
| Reward | 0.916 | 0.917 | 0.934 | 0.9133 | 0.766 | 0.719 |

Table 3: **Wall-Clock Time Comparisons.** Wall-clock time measured MHDD applied to USM guided by ZINC250K QED reward. Batching significantly improves the speed of MHDD, resulting in faster performance than SMC and, at equal batch sizes, roughly as fast as BoN as well.

| | | | ZINC250K | | | | | | | | | | | | |
|---|---|---|---|---|---|---|---|---|---|---|---|---|---|---|---|
| | MHDD | BoN | QED BoN | SMC | SVDD | MHDD | BoN | Rings BoN | SMC | SVDD | MHDD | BoN | SA BoN | SMC | SVDD |
| Time (s) | 334 | 334 | 359 | 359 | 264 | 321 | 331 | 359 | 320 | 276 | 337 | 328 | 300 | 341 | 279 |
| Batch Size | 8 | 78 | 8 | 14 | 886 | 8 | 78 | 8 | 14 | 886 | 8 | 78 | 8 | 14 | 886 |
| NFE | 1024 | 5750 | 1024 | 1024 | 65536 | 1024 | 5750 | 1024 | 1024 | 65536 | 1024 | 5750 | 1024 | 1024 | 65536 |
| Reward | 0.917 | 0.934 | 0.9133 | 0.766 | 0.719 | 5.315 | 4.945 | 4.312 | 2.630 | 2.607 | 0.912 | 0.919 | 0.892 | 0.792 | 0.755 |

Table 4: **ZINC250K Rewards under Matched Wall-Clock Time.** Rewards and time are computed from 128 drawn samples. MHDD (with batching) significantly outperforms all other baselines in ring count while being comparable with BoN in QED and SA.

| | MPRA - HepG2 | | | | |
|---|---|---|---|---|---|
| | MHDD | BoN | BoN | SMC | SVDD |
| Time (s) | 74 | 80 | 103 | 398 | 246 |
| Batch Size | 8 | 79 | 8 | 8 | 8 |
| NFE / Sample | 1000 | 10000 | 2000 | 1000 | 1000 |
| Reward | 4.566 | 3.471 | 2.240 | 0.768 | 0.361 |

Table 5: **MPRA Rewards under Matched Wall-Clock Time.** Rewards and time are computed from 64 drawn samples. MHDD (with batching) significantly outperforms all other baselines in HepG2 activity.

tion with various reward functions demonstrate superior performance of our method compared to previous methods that rely on intermediate rewards.

**Limitations and Future Work.** While our method is effective in reward-guided sampling, using burn-in means many samples are simply discarded. Optimizing the initial sample so that it is closer to the target distribution may help reduce the burn-in time and is left for future work. Lastly, designing or optimizing the proposal distribution that is more aligned with the target distribution is also an exciting area for future research on more efficient sampling.

## ETHICS STATEMENT.

We adhere to the ICLR Code of Ethics. This work uses only publicly available models and datasets and does not involve human subjects, user data, or personally identifiable information. We recognize the potential for misuse of generative AI and encourage responsible deployment of our method.

## REPRODUCIBILITY STATEMENT.

We plan to release the code upon publication. The pseudocode of our method is provided in Alg. 1. Details of the experimental setup are given in App. C. The derivation of the acceptance probability is also included in App. B.

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

APPENDIX

## A  METROPOLIS-HASTINGS ALGORITHM

In this section, we present a brief overview of the Metropolis-Hastings (MH) algorithm. For more details, we refer the reader to chapter 12 section 2 of Murphy (2023).

The MH algorithm constructs a Markov chain which converges to a target distribution $p^\star(x)$. In order to do so, a **proposal distribution** $q(\cdot|x)$ proposes to move from the current state $x$ to a new state $x' \sim q(\cdot|x)$. After proposing the new state $x'$, MH decides whether to accept or reject the new state with the acceptance probability

$$A(x'|x) = \min\left(1, \alpha\right), \quad \alpha = \frac{p^\star(x')q(x|x')}{p^\star(x)q(x'|x)}.$$

Intuitively, the accept-reject step is necessary as the proposal distribution may not match the target distribution. Samples closer to the target distribution are accepted with higher probability whereas samples further from the target distribution are rejected. The MH algorithm is summarized in Alg. 2. The Markov chain constructed by the MH algorithm has the following transition matrix:

$$p(x'|x) = \begin{cases} q(x'|x)A(x'|x) & \text{if } x' \neq x \\ q(x|x) + \sum_{x' \neq x} q(x'|x)(1 - A(x'|x)) & \text{otherwise} \end{cases} \tag{4}$$

---

**Algorithm 2 Metropolis-Hastings Algorithm (MH)**

1: **for** $k = 1, \ldots, K$ **do**
2:     $\tilde{x}^{(k+1)} \sim q(\cdot|x^{(k)})$
3:     $\alpha \leftarrow \frac{p^\star(x')q(x|x')}{p^\star(x)q(x'|x)}$
4:     $A \leftarrow \min(1, \alpha)$
5:     $\rho \sim \mathcal{U}(0, 1)$
6:     $x^{(k+1)} \leftarrow \tilde{x}^{(k+1)}$ if $\rho < A$ else $x^{(k)}$
7: **end for**
8: **return** $\{x^{(1)}, \ldots, x^{(K)}\}$

---

**Theorem A.1** (Theorem 12.2.1 from Murphy (2023)). *If the transition matrix defined by Eq. 4 defined by the MH algorithm is ergodic and irreducible, $p^\star$ is its unique limiting distribution.*

*Proof.* Consider two states $x'$ and $x$. A Markov chain is said to satisfy the detailed balance equation if the following holds:

$$p^\star(x)p(x'|x) = p^\star(x')p(x|x'). \tag{5}$$

It is known that if a Markov chain satisfies the detailed balance equation, then $p^\star$ is its stationary distribution (Theorem 2.6.3 from Murphy (2023)).

To show that $p^\star$ is the unique limiting distribution of the Markov chain defined by Eq. 4, it suffices to show that it satisfies the detailed balance condition in Eq. 5.

Without loss of generality, assume $p^\star(x)q(x'|x) \geq p^\star(x')q(x|x')$. Then, $\alpha = \frac{p^\star(x')q(x|x')}{p^\star(x)q(x'|x)} < 1$ and thus $A(x'|x) = \alpha$. Similarly, by switching the arguments, $A(x|x') = 1$.

To move from $x$ to $x'$, $x'$ must be proposed and accepted. Hence,

$$\begin{aligned} p(x'|x) &= q(x'|x)A(x'|x) && \text{from Eq. 4} \\ &= \frac{p^\star(x')q(x|x')}{p^\star(x)} \end{aligned}$$

It suffices to show that $q(x|x') = p(x|x')$:

$$q(x|x') = q(x|x')A(x|x') \qquad\qquad \because A(x|x') = 1$$
$$= p(x|x') \qquad\qquad\qquad \text{from Eq. 4}$$

Since the MH Markov chain satisfies the detailed balance equation, $p^\star$ is its stationary distribution.

□

## B  DERIVATION OF THE ACCEPTANCE PROBABILITY

Recall from Sec. 4 that the proposal distribution $q(\cdot|x_0)$ is defined in Eq. 2 as:

$$q(x_0'|x_0) := p_t(x_t|x_0)p(x_0'|x_t) \qquad\qquad (6)$$

We calculate $\alpha$ which is used to compute the acceptance probability $A$. First, we draw and fix a noisy sample $x_t \sim p_t(\cdot|x_0)$. Once $x_t$ has been drawn, $\alpha$ simplifies to

$$\alpha = \exp\left(\frac{r(x_0') - r(x_0)}{\beta}\right)\frac{p(x_0')q(x_0|x_0')}{p(x_0)q(x_0'|x_0)}$$
$$= \exp\left(\frac{r(x_0') - r(x_0)}{\beta}\right)\frac{p(x_0')p(x_t|x_0')p(x_0|x_t)}{p(x_0)p(x_t|x_0)p(x_0'|x_t)}$$
$$= \exp\left(\frac{r(x_0') - r(x_0)}{\beta}\right)\frac{p(x_0')p(x_t|x_0')p(x_0|x_t, x_0')}{p(x_0)p(x_t|x_0)p(x_0'|x_t, x_0)}$$
$$= \exp\left(\frac{r(x_0') - r(x_0)}{\beta}\right)\frac{p(x_0, x_t, x_0')}{p(x_0', x_t, x_0)}$$
$$= \exp\left(\frac{r(x_0') - r(x_0)}{\beta}\right),$$

where the third line follows from the conditional independence of $x_0$ and $x_0'$ given $x_t$ due to the Markov property: $p(x_0|x_t) = p(x_0|x_t, x_0')$.

## C  EXPERIMENT DETAILS

### C.1  PRETRAINED MODELS

We provide details on the training setup and hyperparameters of each diffusion model on each dataset in Tab. 6. For QM9 and ZINC250K, we use the transformer architecture for the diffusion, whereas for MPRA we use the CNN architecture, following Stark et al. (2024) and Wang et al. (2025). Note that the SEDD-U model pretrained on MPRA is taken from the publicly available checkpoint provided by Chu et al. (2025). All models except for SEDD-U was trained using Adam (Adam et al., 2014) while SEDD-U was trained using AdamW (Loshchilov & Hutter, 2019).

### C.2  REWARD-GUIDED SAMPLING

We provide experimental details on each evaluation setup. We sample 1024 molecules for QM9 and ZINC250K, and 640 DNA sequences for MPRA. We fix 32 denoising steps for QM9, 74 denoising steps for ZINC250K, and 128 denoising steps for MPRA.

For experiments on the QM9 (Ramakrishnan et al., 2014) and ZINC250K (Irwin et al., 2012) datasets, we fix the total diffusion model NFE per sample to be 1024. For MPRA (Gosai et al., 2023), we fix the NFE as 1000 as done by Chu et al. (2025). Since one run of MHDD generates

| | MDM, USM, and SEDD-M | | | SEDD-U | |
| --- | --- | --- | --- | --- | --- |
| | QM9 | ZINC250K | MPRA | QM9 | ZINC250K |
| Train steps | 25,000 | 50,000 | 131,500 | 40,000 | 100,000 |
| Context size | 32 | 74 | 200 | 32 | 74 |
| Batch size | 1024 | 384 | 512 | 512 | 512 |
| LR | $3e^{-4}$ | $3e^{-4}$ | $2e^{-3}$ | $3e^{-4}$ | $3e^{-4}$ |
| Optim. | ADAM (0.9, 0.999) | ADAM (0.9, 0.999) | ADAM (0.9, 0.999) | ADAMW (0.9, 0.999) | ADAMW (0.9, 0.999) |
| LR sched. | Constant Warmup - | Constant Warmup - | Cosine Decay $3e^{-6}$ min. | Constant Warmup - | Constant Warmup - |
| LR warmup steps | 2,500 | 2,500 | 3,000 | 2,500 | 2,500 |
| GPU count | 2 | 2 | 2 | 8 | 4 |
| GPU type | RTX3090 | RTX3090 | RTX3090 | RTX3090 | RTX3090 |

Table 6: Training setup on QM9, ZINC250K, and MPRA using MDM (Sahoo et al., 2024), USM (Schiff et al., 2024), SEDD-U (Lou et al., 2024), and SEDD-M (Lou et al., 2024).

multiple samples, we draw $S$ samples from the resulting chain by discarding the first half of the chain (burn-in) and taking $S$ equally-spaced samples from the latter half. Since $S$ samples are generated, we scale the NFE by $S$ for MHDD. This scaling highlights one of the key benefits of MHDD: after the initial burn-in period, samples can be generated quickly using a few steps. For all experiments, we fix $\beta = 0.02$. To obtain an $x_0$ prediction from SEDD, we run sampling using one step to jump to time $t = 1$ (clean sample).

Hyperparameters used by our method is shown in Tab. 7. Initial denoising steps refer to the number of steps used to obtain the initial clean sample $x_0^{(1)}$ whereas denoising steps $M$ refer to the number of steps used during the $M$-step reverse process in every MH iteration. Note that $M$ is much smaller than the initial denoising steps.

| | QM9 | ZINC250K | MPRA |
| --- | --- | --- | --- |
| Samples per Iteration $S$ | 128 | 128 | 64 |
| MH Iterations $K$ | 26208 | 26199 | 6390 |
| Initial Denoising Steps | 32 | 74 | 100 |
| Denoising Steps $M$ | 5 | 5 | 10 |
| $t_l$ | 0.5 | 0.5 | 0.5 |
| $t_h$ | 0.8 | 0.8 | 0.8 |

Table 7: Hyperparameters for MHDD.

# D  ADDITIONAL EXPERIMENTAL RESULTS

## D.1  DIVERSITY

We compute the diversity as follows:

- For molecule tasks (QM9 and ZINC250K), we compute the mean pairwise Tanimoto similarity based on the Morgan2 fingerprint, and subtract it from 1.
- For the DNA task (MPRA), compute the mean pairwise cosine similarity of the one-hot encodings, and subtract it from 1.

The results are shown in Tab. 8. BoN and MHDD have a slight decrease in diversity while SMC suffers from significant degradation in diversity. This can be attributed to samples clustering around

the high-reward modes. MHDD consistently achieves a diversity score of over 0.8 on molecule tasks which suggests that the samples are indeed diverse, with an average Tanimoto similarity of less than 0.2. On DNA tasks, the cosine similarity between the sequences is less than 0.3, indicating that the generated sequences are diverse.

| | | QM9 | | | ZINC250K | | | MPRA |
| | | QED | Rings | SA | QED | Rings | SA | HepG2 |
|---|---|---|---|---|---|---|---|---|
| | Pretrained | 0.922 | 0.922 | 0.922 | 0.910 | 0.910 | 0.910 | 0.749 |
| | BoN | 0.904 | 0.884 | 0.894 | 0.876 | 0.874 | 0.876 | 0.749 |
| MDM | SMC | 0.834 | 0.916 | 0.802 | 0.929 | 0.918 | 0.876 | 0.747 |
| | SVDD | 0.920 | 0.920 | 0.920 | 0.900 | 0.896 | 0.900 | 0.747 |
| | **MHDD** | 0.885 | 0.804 | 0.862 | 0.874 | 0.797 | 0.874 | 0.742 |
| | **MHDD-B** | 0.891 | 0.854 | 0.868 | 0.872 | 0.877 | 0.839 | 0.729 |
| | Pretrained | 0.921 | 0.921 | 0.921 | 0.879 | 0.879 | 0.879 | 0.748 |
| | BoN | 0.895 | 0.890 | 0.912 | 0.862 | 0.866 | 0.841 | 0.749 |
| USM | SMC | 0.923 | 0.920 | 0.927 | 0.874 | 0.875 | 0.874 | 0.748 |
| | SVDD | 0.922 | 0.918 | 0.921 | 0.875 | 0.875 | 0.875 | 0.748 |
| | **MHDD** | 0.880 | 0.827 | 0.868 | 0.856 | 0.837 | 0.841 | 0.734 |
| | **MHDD-B** | 0.881 | 0.856 | 0.881 | 0.853 | 0.861 | 0.838 | 0.732 |
| | Pretrained | 0.926 | 0.926 | 0.926 | 0.905 | 0.905 | 0.905 | 0.748 |
| | BoN | 0.903 | 0.889 | 0.926 | 0.874 | 0.874 | 0.877 | 0.749 |
| SEDD-M | SMC | 0.925 | 0.914 | 0.927 | 0.902 | 0.902 | 0.902 | 0.748 |
| | SVDD | 0.928 | 0.928 | 0.928 | 0.907 | 0.907 | 0.907 | 0.748 |
| | **MHDD** | 0.885 | 0.847 | 0.890 | 0.865 | 0.880 | 0.837 | 0.749 |
| | **MHDD-B** | 0.898 | 0.858 | 0.898 | 0.887 | 0.888 | 0.870 | 0.732 |
| | Pretrained | 0.922 | 0.922 | 0.922 | 0.879 | 0.879 | 0.879 | 0.631 |
| | BoN | 0.900 | 0.900 | 0.915 | 0.866 | 0.867 | 0.851 | 0.661 |
| SEDD-U | SMC | 0.901 | 0.905 | 0.910 | 0.872 | 0.873 | 0.866 | 0.661 |
| | SVDD | 0.906 | 0.907 | 0.914 | 0.873 | 0.877 | 0.870 | 0.666 |
| | SGDD | 0.906 | 0.913 | 0.908 | 0.868 | 0.866 | 0.859 | 0.650 |
| | **MHDD** | 0.880 | 0.864 | 0.863 | 0.855 | 0.868 | 0.825 | 0.709 |
| | **MHDD-B** | 0.869 | 0.865 | 0.848 | 0.859 | 0.863 | 0.843 | 0.662 |

Table 8: **Diversity metrics for each method and reward.** Higher is better.

## D.2 NFE COMPARISON

In this section, we compare the performance of each method across various NFEs for QM9 and ZINC250K. We use NFEs $\in \{512, 1024, 2048, 4096\}$ for molecule tasks and NFEs $\in \{500, 1000, 2000, 4000\}$ for MPRA. The results are shown in Fig. 3. As shown in the figure, MHDD consistently and significantly outperforms all other methods in ring count and HepG2 activity across all diffusion models and datasets. MHDD also outperforms all other methods in SA except for BoN when using USM on ZINC250K, and outperforms all other methods in QED except for BoN when using USM. We note that for the hardest reward ring count where the high-reward samples lie in extremely low density regions, MHDD consistently outperforms all other methods by a large margin.

## D.3 TIME PARAMETERS $t_{\mathrm{LO}}$ AND $t_{\mathrm{HI}}$

We provide additional analysis on the time parameters $t_{\mathrm{lo}}$ and $t_{\mathrm{hi}}$. Larger $t_{\mathrm{lo}}$ and $t_{\mathrm{hi}}$ lead to increased exploration as the samples from the proposal distribution become more uncorrelated due to the larger noise scale. Smaller values lead to increased exploitation as samples are more correlated due to the smaller noise scale.

To test the sensitivity of MHDD with respect to these time parameters, we fix $t_{\mathrm{lo}} = 0.2$ and vary $t_{\mathrm{hi}} \in [0.3, 0.8]$. We plot the reward and diversity for ZINC250K USM samples with respect to these parameters. We also include a plot where we fix $t_{\mathrm{hi}} = 0.8$ and vary $t_{\mathrm{lo}} \in [0.2, 0.7]$. The results are shown in Fig. 4. As shown in the figure, our method is not sensitive to the time parameters in general. However, For ring count, the reward decreases as $t_{\mathrm{lo}}$ increases. Intuitively, as $t_{\mathrm{lo}}$ increases, each MHDD step changes the current $x_0$ sample significantly, resulting in greater exploration but less exploitation. For simpler rewards such as QED and SA, this does not significantly impact the

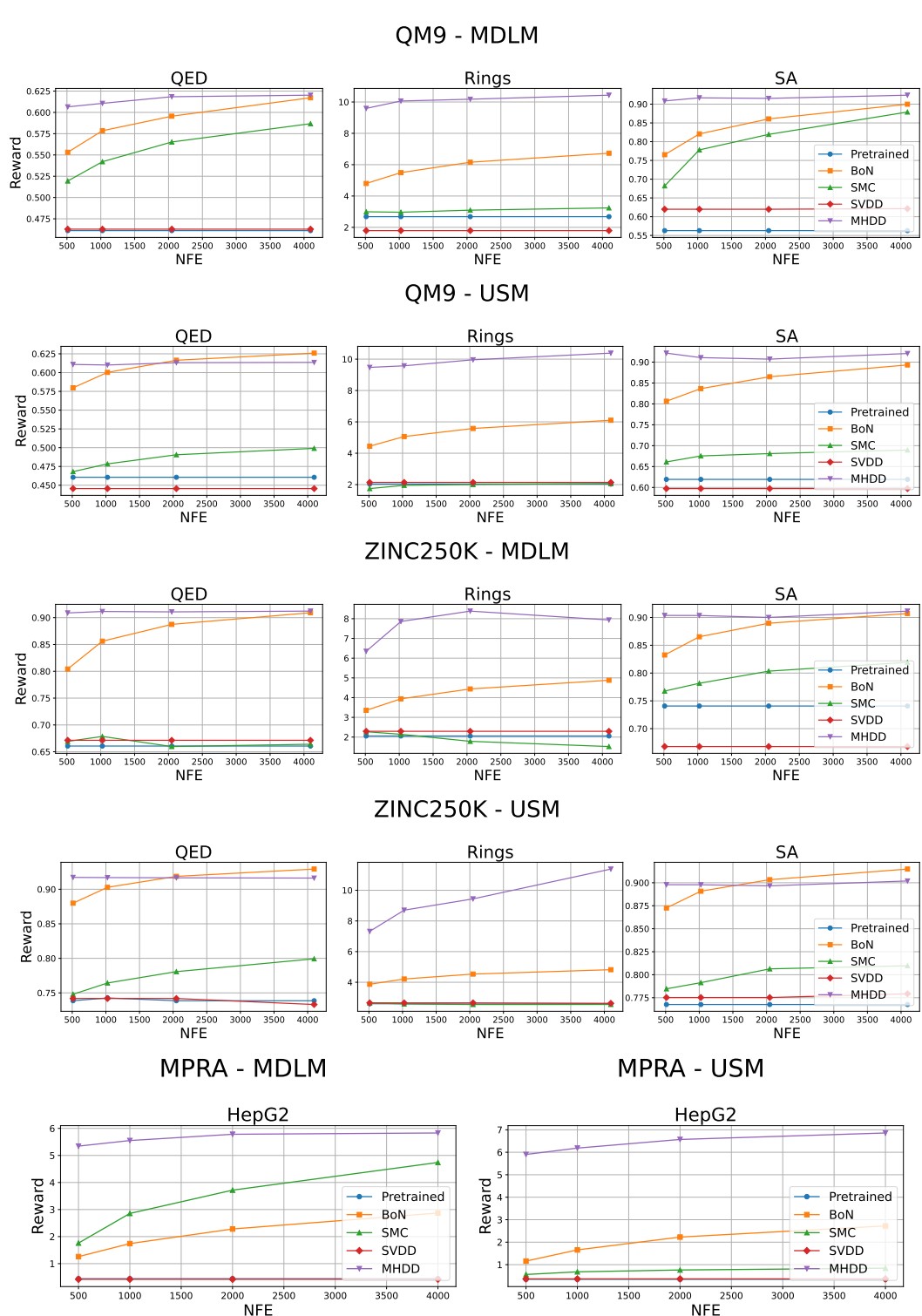

Figure 3: **Performance across various NFEs.** We run each method on molecule generation using NFEs $\in \{512, 1024, 2048, 4096\}$ for molecule tasks and NFE $\in \{500, 1000, 2000, 4000\}$ for MPRA.

average reward. However, for ring count where high-reward samples lie in extremely low density regions of the pretrained model's learned distribution (refer to Fig. 2), it is necessary to choose smaller times to exploit and explore around the current $x_0$.

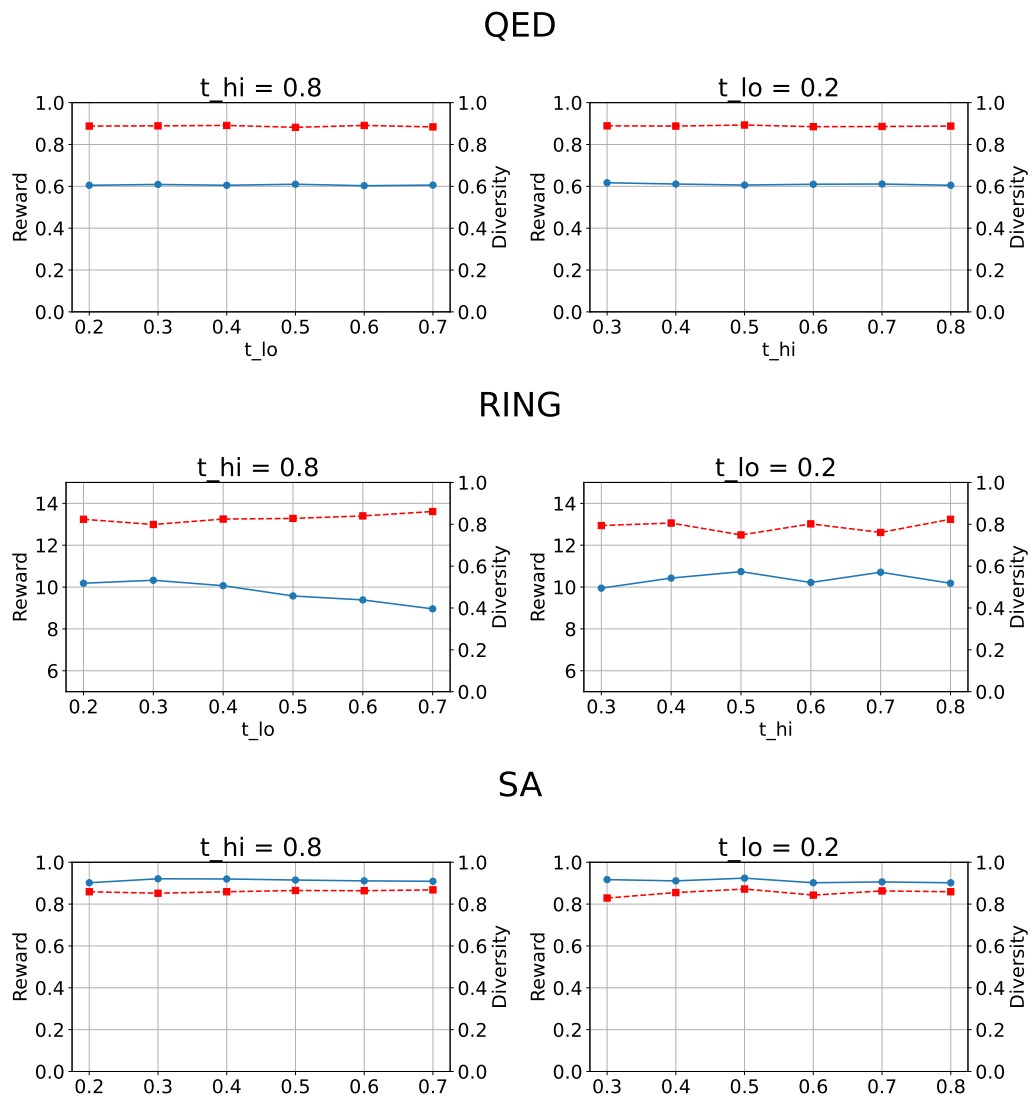

Figure 4: **Analysis of time parameters.** We plot the reward and diversity for ZINC250K USM samples with respect to (1) fixing $t_{\text{lo}} = 0.2$ and vary $t_{\text{hi}} \in [0.3, 0.8]$, and (2) fixing $t_{\text{hi}} = 0.8$ and vary $t_{\text{lo}} \in [0.2, 0.7]$.

### D.4 BURN-IN TIME AND ACCEPTANCE RATE

We provide autocorrelation plots for ZINC250K MDLM and USM in Fig. 5 to further investigate the burn-in time. The autocorrelation function quickly vanishes to zero within the first 2000 iterations and remains small until the end, indicating that the chain converges quickly. Since these metrics are only heuristics for measuring convergence, we conservatively discard the first half of the chain. However, the autocorrelation plots suggest that it may be possible to burn fewer samples.

We also provide acceptance rates for MHDD with MDLM and USM in Tab. 9 as well. Although the acceptance rate for MDLM is quite low due to MDLM often generating invalid molecules, the acceptance rate is still acceptable as we draw 128 samples from 13k samples after the burn-in

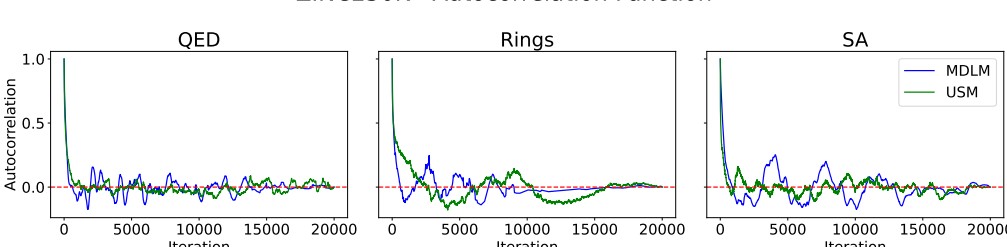

Figure 5: **Autocorrelation Plots for ZINC250K MDLM and USM.**

period. On the other hand, USM has much higher acceptance rates which can be attributed to its higher validity rate due to the ability to correct errors from previous diffusion steps

|  | QM9 | | | ZINC250K | | | MPRA |
|  | QED | Rings | SA | QED | Rings | SA | HepG2 |
|---|---|---|---|---|---|---|---|
| MDM | 0.032 | 0.002 | 0.009 | 0.009 | 0.003 | 0.011 | 0.029 |
| USM | 0.510 | 0.330 | 0.346 | 0.725 | 0.597 | 0.711 | 0.038 |

Table 9: **Acceptance Rates for MDM and USM.**

## E  QUALITATIVE RESULTS

Qualitative results are shown in Fig. 6.

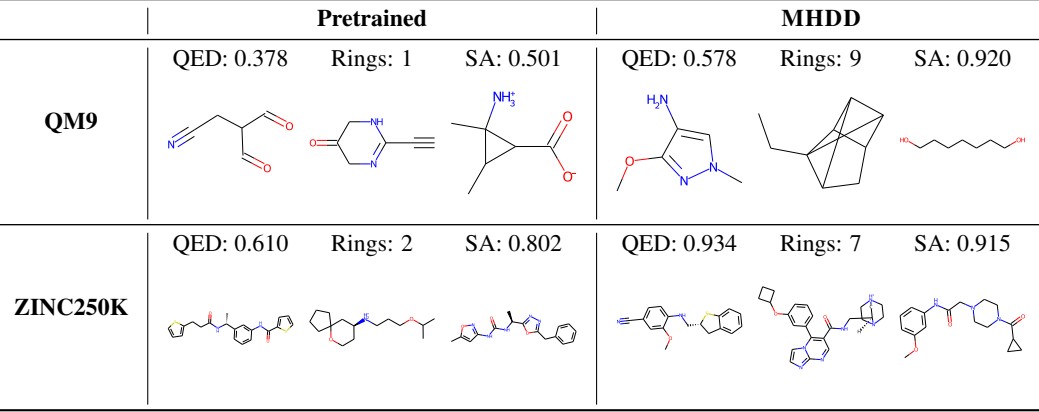

Figure 6: Molecules randomly sampled from discrete diffusion models pretrained on QM9 (Ramakrishnan et al., 2014) and ZINC250K (Irwin et al., 2012). MHDD generates molecules with high rewards as shown on the right.

