# OpenReview forum: "Metropolis-Hastings Discrete Diffusion: Reward-Guided Sampling by Exploring the Clean Data Manifold"
_ICLR.cc/2026/Conference — Submitted to ICLR 2026_

### Official Review · Reviewer_7t7b · 2025-10-29

**Soundness:** 2
**Presentation:** 2
**Contribution:** 2
**Rating:** 2
**Confidence:** 4

**Summary:**

The paper proposes Metropolis–Hastings Discrete Diffusion (MHDD): a training‑free, test‑time sampler that forms a Markov chain over clean samples $x_0$ to optimize a reward‑weighted distribution. At each iteration, MHDD samples a noise level, draws noisy samples $x_t$ via the forward process, runs an M-step reverse from $x_t$ to propose $x_0’$, and accepts with a reward‑only rule.

**Strengths:**

- MHDD applies uniformly across masked vs uniform discrete diffusion and discrete‑time vs CTMC variants with low implementation overhead.
- The paper argues persuasively against noisy intermediate rewards in scientific domains.
- On three datasets and multiple rewards, MHDD tops baselines in mean reward.

**Weaknesses:**

- Eq. (3) is derived in App. B by conditioning on a fixed $x_t$, while Sec. 4.2 defines a forward–backward proposal whose marginal is a mixture over $t$ and $x_t$ (See question 4).
- Table 2 reports point estimates only, no seeds, dispersion, or statistical significance.
- The diagnostics for mixing / convergence are absent. No acceptance rates, burn‑in traces, or sensitivity to $t_{\mathrm{lo}},t_{\mathrm{hi}}$.
- Figure 2 has no legend.
- Code is not yet provided.

**Questions:**

## 1.
Your sampler proposes by partially noising then running an M-step reverse (Table 4 shows M=5 and M=10 for MPRA), which seems intrinsically local in Hamming space. If the chain starts in a low‑reward mode separated by low‑density regions under $p_{\text{pre}}$, it may rarely traverse to a distant high‑reward mode without a very large t (near 1). In other words, MHDD appears to share BoN’s limitation noted in Sec. 4 (lack of trajectory guidance).

Could you argue why MHDD has better mixing properties then BoN? For instance, can you report cross‑mode jump rates, acceptance vs. t, effective sample size, or ablate $t_{\mathrm{lo}},t_{\mathrm{hi}}$?

## 2.
Conceptually, one could use intermediate rewards only to design the proposal (e.g., an SMC‑like local search to generate candidates), then apply an MH correction. Have you explored such “guided‑proposal + MH correction” hybrids?

## 3.
There’s a complementary line of work on discrete diffusion that diffuses in a continuous space and decodes at the end (learned embedding or simplex relaxations). Recent Malliavin‑calculus–based methods allow conditioning on non‑differentiable terminal‑time rewards (Pidstrigach et al., 2025).
Do you expect combining a continuous relaxation with Malliavin‑based terminal guidance to work for your scientific rewards? Can you comment on that approach?

## 4.
In Algorithm 1, the proposal seen by the chain on clean states is the marginal mixture
$ \mathbb E [p_{\mathrm{rev}}(x^\prime_0 \mid x_t)]$ where $p_{\mathrm{rev}}(\cdot\mid x_t)$ is implemented by an M-step reverse process and the expectation is on $t\sim U[t_{\text{lo}},t_{\text{hi}}]$ and $x_t\sim p_t(\cdot | x_0)$. By contrast, Sec. 4.2 writes $q(x^\prime_0\mid x_0):=p_t(x_t\mid x_0)p_{\mathrm{rev}}(x^\prime_0 \mid x_t)$ (Eq. (2)) and App. B derives the reward‑only acceptance $A=\min(1,\exp(\Delta r/\beta))$ (Eq. (3)) by conditioning on a fixed sampled $x_t$ so that the same $x_t$ appears in numerator and denominator.

In a standard MH update on the marginal state space, however, the reverse term involves a fresh draw $x^\prime_t \sim p_t(\cdot\mid x^\prime_0)$, generally different from the $x_t$ used to propose $x^\prime_0$. In that case, the cancellation with $p_{\text{pre}}$ does not automatically follow.

- What is the actual state space of the chain? Is the Markov chain defined on the augmented state $(x_0,t,x_t)$? if so, are $t$ and any reverse‑path randomness included so that the same auxiliaries appear in both directions of the Hastings ratio?
- Under what condition is Eq. (3) valid? Do you assume reversibility of the marginal proposal w.r.t. the pre-trained prior?
- What bias should we expect if the reverse kernel only approximates the posterior?

## 5.
App. C says: “Since S samples are generated, we scale the NFE by S for MHDD” (p. 16), but the exact per‑method accounting is unclear. For MHDD does the number of reverse steps $M$ depend on the sampled $t$? That would make the number of NFEs random.

*Conditioning Diffusions Using Malliavin Calculus*. Jakiw Pidstrigach, Elizabeth L. Baker, Carles Domingo-Enrich, George Deligiannidis, Nikolas Nüsken. (2025). ICML

---

> ### Author Response · Authors · 2025-11-25
> **Response to Reviewer 7t7b, Part 1**
>
> We thank the reviewer for the detailed review. We are pleased that the reviewer finds our observation on the unreliability of intermediate rewards important, and that they find our method effective in multiple datasets and across both uniform and masked discrete diffusion models.
>
> We address all the reviewer's questions and concerns as follows:
> - We provide additional explanation and proof on why it is possible to fix $x_t$ in the derivation of the acceptance probability by showing that the Markov chains with fixed and variable $x_t$ are equivalent. **(W1, Q4)**
> - We include additional diagnostics for mixing and convergence Figure 5 and Table 9in Appendix D.4 of the revision, and perform additional experiments showing that MHDD is not sensitive to choices of $t_\text{lo}$ and $t_\text{hi}$ **(W3)**. Following the reviewer's suggestion, we also include 95% confidence intervals in Table 2. **(W2)**
> - We provide an explanation of why MHDD has better mixing properties than BoN **(Q1)**, and address questions on using guided proposal with MH correction, the applicability of Malliavin calculus [1] to reward-guided sampling, and the number of reverse steps $M$. **(Q3, Q5)**
> - We clarify that figure 2 has no need for a legend **(W4)**, and will release the code upon publication **(W5)**.

---

> ### Author Response · Authors · 2025-11-25
> **Response to Reviewer 7t7b, Part 2**
>
> **W1. Clarification on Derivation of the Acceptance Probability (Eq.(3)).** We thank the reviewer for raising the missing explanation on why it is possible to fix $x_t$ in the derivation of the acceptance probability. We include an explanation on this point in L307-L319 of the revised paper, and we also provide the explanation below.
>
> The state space of the Markov chain is the set of clean samples $x_0$. Rather than augmenting the state to include information about both $t$ and $x_t$, we opt to define the proposals stochastically at each step by first sampling $t \sim \mathcal{U}(t_\text{lo}, t_\text{hi})$ and $x_t \sim p_t(\cdot | x_0)$ and then defining the proposal based on this $t$ and $x_t$. Although this appears to result in a time-inhomogeneous Markov chain, it is actually equivalent to a *time-homogeneous* chain with proposals defined by the marginal mixture $\mathbb{E}[p_\text{rev}(x_0’|x_t)]$ which the reviewer has noted. In particular, we can apply standard convergence proofs involving detailed balance to this time-homogeneous Markov chain to show that MHDD converges to the target distribution. We now provide a brief sketch of the proof of this equivalence.
>
> The time-inhomogeneous chain involves (1) sampling $t$ and $x_t$ first then (2) defining the proposal $q(x’|x, t, x_t)$. However, this two-step process can be combined by considering the probability of drawing each $t$ and $x_t$:
>
> $q(x_0’|x_0) = \int_{t_\text{lo}}^{t_\text{hi}} \int_{\mathbb{x}_t} q(x_0’|x_0, x_t) p(t) p(x_t) dx_t dt = \mathbb{E}\_t \mathbb{E}\_{x_t} [p_t(x_t|x_0)p(x_0’|x_t)]$,
>
> which is time-homogeneous. This is the same proposal $\mathbb{E}[p_\text{rev}(x_0’|x_t)]$ that the reviewer has noted. Since the two Markov chains are equivalent, we can fix $x_t$ at each iteration to make the acceptance probability tractable while still enjoying the theoretical convergence guarantees on the equivalent time-homogeneous chain.
>
> **W2. Table 2 reports point estimates only.** Based on the reviewer's suggestion, we also provide 95% confidence intervals in Table 2 of the revised paper.
>
> **W3. Diagnostics on mixing/convergence.** We thank the reviewer for pointing this out. We include plots of the autocorrelation in Figure 5 in Appendix D.4 of the revision. As shown, the autocorrelation function quickly vanishes to zero within the first 2000 iterations and remains small, suggesting that the chain has mixed and converged. In practice, these metrics are only heuristics, and we conservatively burn the first half of the Markov chain.
>
> We also include additional experiments on the hyperparameters $t_\text{lo}$ and $t_\text{hi}$ in Figure 4 in Appendix D.3 of the revised paper. As shown in the figure, MHDD is not sensitive to $t_\text{lo}$ and $t_\text{hi}$.
>
> **W4. Figure 2 has no legend.** Please note that the titles on each plot in Figure 2 already indicate what each plot represents, and the caption of the Figure describes the interpretation of the plots.
>
> **W5. Code is not yet provided.** As stated in L532 of the paper, we plan to release code upon publication. However, we have also already included the pseudocode in Algorithm 1 as well as other details for reproducibility in Section 5 and Appendix C.

---

> > ### Author Response · Authors · 2025-11-25
> > **Response to Reviewer 7t7b, Part 3**
> >
> > **Q1. Why does MHDD have better mixing properties than BoN?** As explained in L459-L461 BoN independently generates each sample, which results in inefficient exploitation of high-reward samples. In particular, BoN fails to generate high-reward samples (e.g., ring count and HepG2) if such samples lie in extremely low density regions of the pretrained distribution. MHDD, however, balances the exploration-exploitation trade-off via the time parameter $t \in [t_\text{lo}, t_\text{hi}]$. Larger $t$ leads to increased exploration as the samples from the proposal distribution become more uncorrelated due to the larger noise scale, which helps with the mixing property. Smaller $t$ encourages local exploitation around the current sample, which offers trajectory guidance by exploring locally around a sample, unlike BoN which always samples from a different initialization.
> >
> > **Q2. Guided-proposal + MH correction hybrid.** Although applying MH correction together with a reward-guided proposal is possible, it is not aligned with our motivation and methodology. Firstly, the advantage of MHDD comes from utilizing **only clean rewards** instead of noisy intermediate rewards as guidance signals. One of our main contributions is showing that shifting from intermediate to clean rewards significantly improves performance. Secondly, one of our key contributions is using the forward-backward diffusion to model the proposal distribution in order to make the acceptance probability tractable. This is not possible when applying MH correction only to intermediate states, which would require defining a target distribution different from $p_\beta(x_0)$ due to the intractable terms in the acceptance probability. If the reviewer could provide more details on how to apply MH correction using guided-proposals, we will be happy to engage in further discussion.
> >
> > **Q3. Do you expect combining a continuous relaxation with Malliavin‑based terminal guidance to work for your scientific rewards?** In general, we can expect that Malliavin-based terminal guidance [1] combined with continuous diffusion models and a discretization step may also work. However, discrete diffusion still remains the SOTA on discrete data modelling. As such, we can expect that Malliavin-based terminal reward guidance will still perform worse than discrete diffusion. Furthermore, Malliavin-based terminal guidance is training-based and will incur additional training costs for each reward function, and the experiments conducted in [1] are small-scale and may not sufficiently scale for scientific applications.
> >
> > **Q4. Derivation of the Acceptance Probability.** Please refer to our response to W1.
> >
> > **Q5. Does the number of reverse steps M depend on the sampled t?** No, the number of reverse steps M is fixed regardless of the value of the sampled t. The number of iterations of MHDD is then computed based on this fixed M: for an NFE of 1024 per sample with K=32, M=5, and 128 samples per chain, MHDD uses (1024\*128-32)/5=26208 iterations (QM9 in Table 6 of the Appendix C).
> >
> > ---
> > [1] Pidstrigach, Jakiw, et al. "Conditioning Diffusions Using Malliavin Calculus." _Forty-second International Conference on Machine Learning_.

---

### Official Review · Reviewer_1zUF · 2025-10-29

**Soundness:** 3
**Presentation:** 3
**Contribution:** 2
**Rating:** 4
**Confidence:** 4

**Summary:**

This paper suggests a method to generate reward-oriented samples. Compared with previous baselines, authors suggest to only rely on clean sample reward, getting target distribution by metropolis hasting.

**Strengths:**

The idea is interesting, since the authors point out that on some scientific cases, process reward model is easy to make mistakes.
The suggested MHDD is easy to implement but effective.

**Weaknesses:**

- Convergence: while metropolis hastings can converge to some distributions in the end, it’s unclear that MHDD will finally converge to the target distribution $p_{\beta}(x_0)$. The chain may finally converge to a bad distribution or remain confined to limited regions.
- Efficiency: The claimed efficiency improvement over Best-of-N sampling is not clearly demonstrated under compute-matched conditions. The authors claim that the method is compared under the same model NFE. To me, it should be a curve where x-axis is the computation and y-axis is the performance. Could the authors show this kind of diagram so that readers know the performance comparison under different computation resources?
- Initialization: The method may heavily depend on the choice of the initial sample $x_0$; poor initialization could lead to slow convergence rate or suboptimal results. In this way, authors should repeat the experiments many times to have the error bar. Authors can also consider (not mandatory) compare the differences when sampling good $x_0$, bad $x_0$, or random $x_0$.
- Diversity: Since proposals are local noise–denoise perturbations, generated samples might stay close to the starting region, potentially limiting coverage and sample diversity. Could the authors show the divergence measure, on generated samples, compared with baselines?
Uniform diffusion:

**Questions:**

- Could SGDD experiment on USM? Authors say SGDD is based on uniform diffusion, so only experiment it on SEDD-U, but how about USM?
- SMC and SVDD seem to fail at all experimental datasets, which is very different from my sense but still interesting. Since those are actually effective methods in many cases, do the authors have any ideas why they can be so worse than BoN?

---

> ### Author Response · Authors · 2025-11-25
> **Response to Reviewer 1zUF, Part 1**
>
> We thank the reviewer for the suggestions and insightful comments. We are pleased that the reviewer finds our observation on the noisy process reward and our solution using MHDD interesting. We are also pleased that they find MHDD effective and easy to implement.
>
> We address all the reviewer's comments and concerns as follows:
> - We address concerns about the efficiency of MHDD by plotting reward against NFE and showing that MHDD consistently outperforms other baselines. **We also additional modify MHDD to support batching, resulting in MHDD-B.** This simple modification significantly reduces the wall-clock time of MHDD with only marginal degradation in performance. **(W2)**
> - We additionally report the diversity of MHDD samples in Table 8 in Appendix D.1 of the revised paper **(W4)** , and additionally include 95% confidence intervals in Table 2 **(W3)**.
> - We clarify that MHDD inherit the convergence properties of the Metropolis-Hastings (as explained in Appendix A) **(W1)**, and that the results in Table 2 are obtained by running multiple MHDD chains with different initializations **(W3)**
> - We clarify that SGDD was specifically tailored to CTMC, not discrete-time, diffusion models and include an explanation of why SMC and SVDD suffer from degraded performance. **(Q1, Q2)**

---

> ### Author Response · Authors · 2025-11-25
> **Response to Reviewer 1zUF, Part 2**
>
> **W1. Convergence: It is unclear that MHDD will finally converge to the target distribution.** As we have explained in L278-L288 and Appendix A, Metropolis-Hastings will eventually converge to the **target** distribution. For proofs of convergence, please refer to Appendix A of the paper. Since MHDD employs Metropolis-Hastings, this same convergence property holds as well. If any clarifications are needed, we would be happy to provide further clarifications.
>
> **W2. Efficiency: Performance comparison under different computation resources.** We additionally include a plot of the average reward across different NFEs in Figure 3 in Appendix D.2. As shown in the Figure, MHDD consistently outperforms all other methods in ring count and HepG2 activity across MDM and USM and datasets. MHDD also outperforms all other methods in SA except for BoN when using USM on ZINC250K, and outperforms all other methods in QED except for BoN when using USM. We note that for the hardest rewards. ring count  and HepG2, where the high-reward samples lie in extremely low density regions, MHDD consistently outperforms all other methods by a large margin.
>
> Additionally, we included results when using MHDD with batching: we run multiple MHDD chains in parallel while decreasing the length of each chain to keep the total NFE unchanged. We refer to this method as **MHDD-B**. We provide a comparison of the performance of MHDD-B and other methods under equal wall-clock time in the table below. We ensure that MHDD-B uses less wall-clock time than other methods except for SVDD which is efficient even at high NFEs but performs poorly. As shown, MHDD-B significantly outperforms all other baselines in ring count and HepG2 rewards while being comparable with BoN in QED and SA. When we match the batch sizes of MHDD-B and BoN, MHDD-B outperforms BoN in all rewards.
>
> | | | QED   |  |  |  |  |   |   | Ring  |  |  |  |
> | - | - | - | - | - | - | - | - | - | - | - | - | - |
> |  | MHDD-B | BoN   | BoN    | SMC   | SVDD  |  |  | MHDD-B | BoN   | BoN   | SMC   | SVDD  |
> | Time (s) | 334    | 334   | 359    | 359   | 264   |  | Time (s)     | 321    | 331   | 359   | 320   | 276 |
> | Batch Size   | 8      | 78    | 8      | 14    | 886   |  | Batch Size   | 8      | 78    | 8     | 14    | 886   |
> | NFE / Sample | 1024   | 5750  | 1024   | 1024  | 65536 |  | NFE / Sample | 1024   | 5750  | 1024  | 1024  | 65536 |
> | Reward  | 0.917  | 0.934 | 0.9133 | 0.766 | 0.719 |     | Reward       | 5.315  | 4.945 | 4.312 | 2.630 | 2.607 |
>
> |   |   | SA    | |  |   |     |    |  | HepG2 |  |  |  |
> | - | - | - | - | - | - | - | - | - | - | - | -| - |
> |              | MHDD-B | BoN   | BoN   | SMC   | SVDD  |  |   | MHDD-B | BoN   | BoN   | SMC   | SVDD  |
> | Time (s)     | 337    | 328   | 300   | 341   | 279   |   | Time (s)     | 74     | 80    | 103   | 398   | 246   |
> | Batch Size   | 8      | 78    | 8     | 14    | 886   |   | Batch Size   | 8      | 79    | 8     | 8     | 8     |
> | NFE / Sample | 1024   | 5750  | 1024  | 1024  | 65536 |  | NFE / Sample | 1000   | 10000 | 2000  | 1000  | 1000  |
> | Reward       | 0.912  | 0.919 | 0.892 | 0.792 | 0.755 |   | Reward    | 4.566  | 3.471 | 2.240 | 0.768 | 0.361 |
>
> **W3. Initialization: The method may heavily depend on the choice of the initial sample.** Our method is not sensitive to the initialization. We clarify that the samples generated in each experiment are taken from multiple Markov chains, each with a different initial sample. For instance, in the molecule experiments, the reported rewards are computed over 1024 samples. Since each chain only produces 128 samples, we run eight chains with different seeds to produce 1024 samples. Based on the reviewer's suggestion, we also added 95% confidence intervals in Table 2 of the revised paper. In the original paper, results for SMC and SVDD for SEDD-U were taken from [3]. However, to obtain the confidence intervals, we rerun the experiments and report the new values in Table 2.
>
> In general our method is not sensitive to initialization as we run the MHDD sufficiently long so that the chain converges. We further include autocorrelation plots in Figure 5 of Appendix D.4 indication that MHDD converges quickly within the first 2000 iterations. Moreover, when using MHDD-B (please refer to our response to W2 for more details on MHDD-B), different initializations are used for each chain in the batch, further mitigating sensitivity to the initialization.
>
> **W4. Diversity: Does MHDD suffer from limited diversity?** We thank the reviewer for raising this important point. We additionally include diversity metrics in Table 8 in Appendix D.1 of the revised paper. As shown in the table, MHDD still produces sufficiently diverse samples, although its diversity is lower than the other methods due to sampling closer to high-reward modes. In general, we can expect a trade-off between reward maximization and diversity, and the overall diversity of the MHDD samples is still sufficiently high.

---

> > ### Author Response · Authors · 2025-11-25
> > **Response to Reviewer 1zUF, Part 3**
> >
> > **Q1. Does SGDD apply to USM?** Please note that SGDD was derived based on characteristics of the uniform CTMC forward process. As such, SGDD is not directly applicable to USM which uses a discrete-time formulation. We have made this point clearer in L392-L393 of the revision.
> >
> > **Q2. Why are SMC and SVDD worse than BoN?** As explained in L427, SMC and SVDD both utilize intermediate rewards as guidance signals. Although this may work well in continuous diffusion where the $x_0$-prediction has a meaningful interpretation via Tweedies’ formula [1,2], the same interpretation does not hold for discrete diffusion. Furthermore, the applications we considered are cases where intermediate rewards are noisy and do not provide meaningful signal for guidance, resulting in worse performance for SMC and SVDD. Please refer to L427-L431 for a more detailed discussion.
> >
> > ---
> > [1] Efron, Bradley. "Defining the curvature of a statistical problem (with applications to second order efficiency)." The Annals of Statistics (1975): 1189-1242.
> >
> > [2] Chung, Hyungjin, et al. "Diffusion Posterior Sampling for General Noisy Inverse Problems." The Eleventh International Conference on Learning Representations.
> >
> > [3] Chu, Wenda, et al. "Split gibbs discrete diffusion posterior sampling." The Thirty-Ninth Annual Conference on Neural Information Processing Systems (2025).

---

> > > ### Comment · Reviewer_1zUF · 2025-11-28
> > >
> > > First, I have to thank the authors for their hard work!!
> > >
> > > The most interesting aspect of this work is, based on a well-known noise–denoise diffusion iteration, the authors introduce an MH procedure and demonstrate convergence. This is conceptually interesting.
> > >
> > > However, even after reviewing the newly added experiments, I still have several fundamental concerns. I am not requesting additional experiments; I simply hope to have a technical conversation/discussion. If the authors find any of my responses are not correct, don’t hesitate and correct me.
> > >
> > > 1. **Discussion on why MHDD can outperform BoN, and why convergence rate heavily depends on the choice of $x_0$​ and $t_0$​**
> > >
> > > From Algorithm 1, MHDD first samples an initial state $t_0$​, then performs a full denoising process to obtain a proposal $x_0'$​.
> > >
> > > In the extreme case where $t_0 = t_T$​, i.e., noise into the maximally noised state (e.g., all-[MASK] in masked diffusion), MHDD essentially reduces to BoN. Each proposal is simply an independent sample from the original diffusion model. (correct me if this is wrong)
> > >
> > > The reason MHDD can outperform BoN is that, in practice, $t_0$​ is often sampled from an intermediate diffusion state, which carries partial information from a previously accepted $x_0$​. This warm-start effect makes subsequent proposals more likely to remain near high-reward regions. If this understanding is correct, then the quality of the currently accepted $x_0$​ is important: a good $x_0$​ accelerates acceleration, while a bad $x_0$​ may slow the process (and if $t_0$​ is close to $t_T$, the effect vanishes) (note I’m not saying converge or not, I’m talking about convergence rate)
> > >
> > > From this perspective, MH guarantees eventual convergence, but the practical efficiency and the advantage over BoN appears to come from these warm-start intermediate states rather than the MH rule.
> > >
> > > 2. **On the claimed limitations of intermediate-value–guided methods such as SMC and SVDD**
> > >
> > > Let me restate my understanding using a concrete example. Suppose at state $x_t$​, the final denoising distribution is:
> > > - 70% probability → $x_{0,a}$, reward 0
> > > - 30% probability → $x_{0,b}$​, reward 10
> > >
> > > The paper argues that SVDD may estimate $v(x_t)=0$ because it performs an argmax decoding and yields $x_{0,a}$​, ignoring the potential high-reward trajectory toward $x_{0,b}$​.
> > >
> > > While this diagnosis is plausible, I believe the issue is not the intermediate value estimation, but the argmax approximation used to obtain it. If one performs multiple temperature-based samples from $p(x_0|x_t)$, people can obtain an unbiased estimate $v(x_t)=3$. The problem arises from selecting the argmax, not from the intermediate value.
> > >
> > > Regarding MHDD: in the above example, MHDD still relies on a (although M-step) full denoising run to obtain the proposal $x_0'$​. It is still more likely to get $x_{0,a}$​ than $x_{0,b}$​. Running multiple proposals helps, but SVDD could also adopt the same strategy. Thus I do not yet see how MHDD fundamentally mitigates the failure mode attributed to intermediate-value–guided methods.
> > >
> > > The above two questions are not about "works", but on "why it works".
> > >
> > > A final quick question:
> > >
> > > > Based on the reviewer's suggestion, we also added 95% confidence intervals in Table 2 of the revised paper. In the original paper, results for SMC and SVDD for SEDD-U were taken from [3].
> > >
> > > I didn't find the results in [3], could you give me the specific tables or page number? In case of different version of paper, could you also share the link?
> > >
> > > Thanks again for your reply!

---

> ### Author Response · Authors · 2025-11-29
>
> We thank the reviewer for the active discussion and interest in our work. We address the additional questions and concerns below.
>
> **Q1. Discussion on why MHDD can outperform BoN, and why convergence rate heavily depends on the choice of $x_0$ and $t_0$.**
> The reviewer is correct that MHDD outperforms BoN due to $t_0$ carrying partial information from the previous $x_0$. Our design choice involves sampling $t_0 \sim [t_\text{lo}, t_\text{hi}]$, enabling the sampling of larger $t_0$ as well (we choose $t_\text{lo}=0.5$ and $t_\text{hi}=0.8$ for our main experiments). Larger $t_0$ leads to increased exploration, which helps convergence by preventing the process from being stuck in local minima. However, larger sampled $t_0$ may lead to potentially bad $x_0'$ proposals due to increased exploration. The MH correction step is necessary to ensure that such bad proposals are rejected, allowing us to sample from a large range of $t_0$ while being more efficient than BoN. In practice, we run MHDD sufficiently long so that even a more limited range of $t_0$ can still result in sufficient exploration (applying multiple local changes successively can still steer the $x_0$ to a different region), which results in MHDD's insensitivity to the choice of $[t_\text{lo}, t_\text{hi}]$ as shown in Appendix D.3.
>
> Finally, we would like to clarify that BoN is MHDD with $t_0=T$ and $\beta \to 0$ (which would result in the MH probability being 1 if the current reward is higher than the previous reward and 0 otherwise). The reviewer is still correct that this special case (even without $\beta \to 0$) inherits all the inefficiencies from BoN, but using $\beta>0$ would still result in a different algorithm which converges to a different stationary distribution than the BoN-induced sampling distribution.
>
> **Q2. On the claimed limitations of intermediate-value–guided methods such as SMC and SVDD.**
> We clarify that by intermediate rewards, we do not mean the reward of the *final* $x_0$ but rather the reward of the $x_0$ prediction $\hat{x}_0(x_t)$ computed at intermediate noisy samples $x_t$ (as described in L246-247 and L258-259). This $\hat{x}_0(x_t)$ is not the true final samples but is the neural network prediction of possible $x_0$ based on $x_t$, as described in L197. Importantly, $\hat{x}_0(x_t)$ is noisy and does not accurately represent the true $x_0$, especially during earlier timesteps. For example, many tokens of $\hat{x}_0(x_t)$ are usually different from that of $x_0$.  In scientific applications, a perturbation of only one or two characters can completely change the reward to zero as shown in Figure 1 (Left) of our paper. As a result, this noisy and inaccurate intermediate rewards can mislead SMC and SVDD since it does not accurately represent the distribution of the true rewards.
>
> For example, consider the reviewer's example with **final** reward distribution
> - $r(x_{0,a})=0$ with 70% probability
> - $r(x_{0,b})=10$ with 30% probability
>
> This final reward distribution may have the following **intermediate** reward distribution
> - $r(\hat{x}_0(x_t)_a)=0$ with 95% probability
> - $r(\hat{x}_0(x_t)_b)=10$ with 5% probability.
>
> Although $x_t$ has a *true* expected reward of 3, the *intermediate* expected reward is 0.5, which is significantly lower. In practice, this is often the case in our scientific applications as the $x_0$ predictions are inaccurate and often result in invalid molecules (which has zero reward) with high probability.
>
> SMC and SVDD may also be applied by running a full denoising run at each step to approximate the final reward distribution, but this results in significant computation overhead: with only 10 particles and a 32-step diffusion process, a 5-step $x_0$ prediction would require 50 NFEs per iteration, resulting in 1600 NFEs per sample. The same setting in MHDD yields 313 clean samples, which is significantly more efficient.
>
> To summarize, MHDD mitigates (1) the inaccuracy of intermediate rewards by leveraging only clean rewards while (2) using the MH algorithm for efficient sampling.
>
> **Q3. Results in [3]**
> Thank you for pointing this out. In the original paper, the results for SMC and SVDD for SEDD-U MPRA were taken from Table 3 (SMC and SVDD-PM) of [3] (https://arxiv.org/abs/2503.01161). However, not all the hyperparameters (number of reverse steps in SVDD for example) used for SMC and SVDD in [3] are  publicly available so the results could not be perfectly replicated. We have updated the values in the paper with our replicated results and error bars.

---

### Official Review · Reviewer_gv3v · 2025-10-30

**Soundness:** 3
**Presentation:** 3
**Contribution:** 2
**Rating:** 6
**Confidence:** 4

**Summary:**

This paper focuses on the challenge of reward-guided sampling for discrete diffusion models in scientific domains (e.g., chemistry and biology), where discrete diffusion models have shown promise in generating discrete data such as molecules and DNA sequences. However, existing reward-guided approaches face critical limitations: gradient-based methods designed for continuous diffusion are inapplicable to discrete spaces, while methods relying on intermediate rewards suffer from noise due to the non-smooth nature of reward functions in scientific fields (e.g., a single character change in a SMILES string can invalidate a molecule and collapse its reward to zero).

To address these issues, the authors propose Metropolis-Hastings Discrete Diffusion (MHDD), a training-free method that enables effective reward-guided sampling without relying on intermediate rewards. The core idea is to construct a Markov chain of clean samples using the Metropolis-Hastings algorithm, with the target reward-weighted distribution as its stationary distribution. A key technical innovation is the design of a proposal distribution via sequential forward (corrupting clean samples) and backward (denoising noisy samples) processes, which makes the otherwise intractable acceptance probability in the Metropolis-Hastings algorithm tractable.

Experiments on molecule generation (QM9, ZINC250K datasets) with three reward functions (QED for drug-likeness, ring count, synthetic accessibility SA) and biological sequence generation (MPRA dataset) with HepG2 enhancer activity reward demonstrate that MHDD consistently outperforms prior methods (e.g., Best-of-N, SMC, SVDD, SGDD) across all discrete diffusion frameworks (masked diffusion MDM, uniform state diffusion USM, score-based CTMC diffusion SEDD-M/SEDD-U).

**Strengths:**

The major contribution is that this paper breaks the reliance of reward-guided sampling in discrete diffusion on intermediate rewards. It combines the Metropolis-Hastings algorithm with the forward-backward processes of discrete diffusion to construct a Markov chain of clean samples. The design of the forward-backward proposal distribution cleverly solves the intractable acceptance probability problem when applying Metropolis-Hastings to diffusion models, presenting novelty in terms of methodology.

**Weaknesses:**

1.	Low Burn-In Efficiency and Unquantified Time Overhead: Requires discarding the first half of the Markov chain (burn-in period) to ensure convergence. For example, in some datasets, when the total number of iterations K is 6390, 3195 iterations are discarded during burn-in. However, the paper does not quantify the additional time overhead caused by the burn-in period and K iterations. Time is a critical indicator in practical applications such as rapid drug screening; the lack of this information makes it impossible to evaluate the practical efficiency of the method and compare its time competitiveness with methods like Best-of-N (BoN) and SMC.
2.	Fixed Proposal Distribution Hyperparameters and No Time Sensitivity Analysis: Relies on manually set hyperparameters (e.g., tlo=0.5, thi=0.8, M=5/10) for the proposal distribution. There is no analysis of how these hyperparameters affect sampling time—for instance, whether increasing M significantly prolongs the time per iteration, or if adjusting tlo/thi can shorten time while maintaining reward performance. Additionally, there is no adaptive hyperparameter adjustment strategy, resulting in insufficient flexibility and a lack of exploration into the trade-off between time and performance.
3.	Lack of Time Overhead Comparison with Other Methods: Although the total NFE is controlled to ensure fairness in experiments, NFE is not entirely equivalent to actual time overhead (the computational complexity of single-step NFE may vary across methods). No actual time consumption comparison between MHDD and methods like BoN, SMC, and SVDD is provided, making it impossible to fully judge MHDD's comprehensive advantages in the "performance-time" dimension.

**Questions:**

1.	The paper notes that the burn-in period requires discarding the first half of the Markov chain (e.g., 3195 iterations discarded when K=6390 for certain datasets) to ensure convergence, but it does not quantify the additional time overhead caused by the burn-in period and K iterations. Given that time is critical for practical applications like rapid drug screening, can you supplement experimental data to quantify the time consumed by the burn-in period and K iterations? Additionally, can you compare MHDD's total time overhead with that of methods like Best-of-N (BoN) and SMC in the same experimental tasks to clarify its time competitiveness?
2.	The proposal distribution of MHDD relies on manually set hyperparameters such as tlo=0.5, thi=0.8, and M=5/10, yet there is no analysis of how these hyperparameters impact sampling time. For example, does increasing M significantly extend the time per iteration? Can adjusting tlo or thi reduce sampling time while maintaining reward performance? Moreover, since there is no adaptive hyperparameter adjustment strategy currently, do you have plans to explore such strategies to balance time overhead and performance flexibility?
3.	While the paper controls the total Number of Function Evaluations (NFE) to ensure experimental fairness, NFE does not fully equate to actual time overhead—single-step NFE may differ in computational complexity across methods (e.g., MHDD's forward-backward process vs. SMC's particle resampling). Can you provide specific data on the actual time consumption of MHDD versus methods like BoN, SMC, and SVDD under the same total NFE? This would help fully assess MHDD's comprehensive advantages in the "performance-time" dimension.

---

> ### Author Response · Authors · 2025-11-25
> **Response to Reviewer gv3v, Part 1**
>
> We thank the reviewer for their detailed and insightful review. We are pleased that the reviewer finds our focus on clean-reward guidance important and that our design of the forward-backward proposal distribution cleverly solves the intractable acceptance probability problem when applying Metropolis-Hastings to diffusion models.
>
> We respond to all the reviewer's concerns and questions as follows:
> - To address burn-in efficiency and time overhead, **we propose batching MHDD** by running multiple chains in parallel and decreasing the length of each chain. We refer to this batched version of MHDD as **MHDD-B**. This simple modification significantly improves the time overhead with minimal degradation in performance.  **(W1, Q1)**
> - We include additional experiments showing that MHDD is insensitive to the hyperparamaters by running and including additional hyperparameter studies. **(W2, Q2)** We also report the wall-clock time of **MHDD-B** (batched MHDD) and include additional comparisons of MHDD-B with other baselines under equal wall-clock time, showing the effectiveness of our method under equal computation time budget **(W3, Q3)**.

---

> ### Author Response · Authors · 2025-11-25
> **Response to Reviewer gv3v, Part 2**
>
> **W1. Burn-in efficiency and time overhead.** We thank the reviewer for raising this point. The burn-in period and $K$ iterations combined costs \~6 minutes for MPRA and \~25 minutes for the molecule tasks. **However, we can speed up MHDD further by introducing batching: we run multiple chains in parallel and decrease the length of each chain accordingly: we call this MHDD-B.** This simple modification significantly improves the wall-clock time with minimal degradation in performance. For example, using a batch size of 4 for MPRA and 8 for molecule tasks, the burn-in time drops to **~3 minutes for MPRA and ~2 minutes for molecule tasks**. We add these new results in Table 2 and L397 of the revised paper, and include a more detailed explanation of MHDD-B. We also include wall-clock time comparisons in our response to Q3.
>
> **W2. Proposal Distribution Hyperparameters.** First, we would like to clarify that $M$, $t_\text{lo}$, and $t_\text{hi}$ do not affect the total time of the algorithm as the NFE is fixed. However, the time spent per iteration increases linearly with $M$ (but is not affected by $t_\text{lo}$, and $t_\text{hi}$ since we fix the number of diffusion steps $M$ regardless of the value of $t \in [t_\text{lo}, t_\text{hi}]$). In general, MHDD is not sensitive to values of $M$ as shown below, except for $M=20$ which significantly decreases the number of MHDD iterations and may result in non-convergence. However, the table below shows that using $M=5$ results in optimal performance in terms of reward while also giving the fastest time per iteration. We also further include analysis on the choice of $t_{lo}$ and $t_{hi}$ in Appendix D.3, showing that MHDD is not sensitive to the choice of $t_{lo}$ and $t_{hi}$.
>
> |     |       | QM9   |       |       | ZINC250K  |       |
> | --- | ----- | ----- | ----- | ----- | --------- | ----- |
> | M   | QED   | Ring  | SA    | QED   | Ring      | SA    |
> | 5   | 0.610 | 9.574 | 0.911 | 0.917 |     8.703 | 0.898 |
> | 10  | 0.601 | 9.392 | 0.913 | 0.910 |     8.477 | 0.901 |
> | 15  | 0.609 | 9.582 | 0.904 | 0.917 |     8.256 | 0.895 |
> | 20  | 0.601 | 9.062 | 0.919 | 0.917 |     6.885 | 0.897 |
>
> **W3. Lack of Time Overhead Comparison with Other Methods.** We thank the reviewer for raising the importance of performance (wall-clock) time. Since MHDD is sequential, the wall-clock time scales linearly with the number of iterations. To remove this limitation, **we propose MHDD-B, a batched version of MHDD in this revision**. In MHDD-B, we run multiple MHDD chains in parallel while decreasing the length of each chain to keep the total NFE unchanged. Additional results for MHDD-B are added in Table 2 and L397 of the revised paper. This simple modification significantly speeds up our method without much degradation in performance. We include a comparison of wall-clock time for ZINC250K QED using USM in the table below. As shown, using MHDD-B speeds up MHDD by a factor of ~9x, with run times faster than SMC, and comparable to BoN when using the same batch size of 8. Furthermore, MHDD-B achieves the best performance among all baselines in the table below.
>
> |            | MHDD  | MHDD-B | BoN   | BoN   | SMC   | SVDD  |
> | ---------- | ----- | ------ | ----- | ----- | ----- | ----- |
> | Time (s)   | 3029  | 334    | 138   | 316   | 359   | 264   |
> | Batch Size | 1     |     8  |   14  |   8   |   14  | 14    |
> | NFE/Sample | 1024  | 1024   | 1024  | 1024  | 1024  | 1024  |
> | Reward     | 0.916 | 0.917  | 0.910 | 0.913 | 0.766 | 0.719 |
>
> **Q1. Time overhead caused by burn-in and K iterations.** Please refer to our response to W1 for details on the time overhead. We also provide specific data on the wall-clock time for MHDD and MHDD-B in our response to Q3.
>
> **Q2. Analysis of Hyperparameters.** Please refer to our response to W2.
>
> **Q3. Actual time consumption of MHDD.** We provide the wall-clock time comparisons for generating 128 samples using NFE=1024 per sample in the table below. As shown, our method is faster than SMC, and comparable to BoN in speed when using the same batch size of 8. Please refer to our response to W3 for more details and discussion on MHDD-B.
>
> |            | MHDD  | MHDD-B | BoN   | BoN   | SMC   | SVDD  |
> | ---------- | ----- | ------ | ----- | ----- | ----- | ----- |
> | Time (s)   | 3029  | 334    | 138   | 316   | 359   | 264   |
> | Batch Size | 1     |     8  |   14  |   8   |   14  | 14    |
> | NFE/Sample | 1024  | 1024   | 1024  | 1024  | 1024  | 1024  |
> | Reward     | 0.916 | 0.917  | 0.910 | 0.913 | 0.766 | 0.719 |

---

### Official Review · Reviewer_a6b4 · 2025-11-01

**Soundness:** 2
**Presentation:** 2
**Contribution:** 2
**Rating:** 2
**Confidence:** 4

**Summary:**

The paper introduces MHDD, a reward-guided sampler for discrete diffusion models that avoids noisy intermediate rewards by doing Metropolis-Hastings on the clean data manifold. The key idea is a forward-backward, starting from a clean sample, applying the forward corruption to a random time, then running a short reverse denoising to propose a new clean sample. This makes MH acceptance ratio tractable and (under their construction) depend on reward differences. Experiments on molecules and DNA sequence show improvements over baselines. The method is claimed to be generally applicable to masked and uniform discrete diffusion and to deliver stronger guidance when process rewards are unreliable.

**Strengths:**

1. By constructing proposals with forward–reverse diffusion, the acceptance rule eliminates intractable terms and depends on rewards. This motivation matches domain facts.

2. Model-agnostic across discrete diffusion families. makes the work broader than SGDD, which targets uniform settings.

3. Simple acceptance rule and reusable chain can be computationally light useful for batch design.

**Weaknesses:**

1. The acceptance simplification assumes proposal factors that make the MH ratio tractable. In practice distribution is approximated by a learned reverse model. This introduces modeling bias that can break detailed balance and stationarity guarantees. Analyses quantifying bias under approximate reverse kernels (e.g., pseudo-marginal MH or noisy MH analyses) and diagnostics of reversibility violations can make claims more grounded.

2. Where did p_{pre} go in acceptance? The paper defines the target but ends with an acceptance depending only on r. It's questionable whether the proposal construction implicitly incorporates the prior so it cancels, and under which conditions. Explicit derivation showing cancellation with learned reverse dynamics, and an ablation demonstrating that removing the forward–backward construction changes the implied target (e.g., acceptance starts to need p_{pre} terms) are needed.

3. Mixing/burn-in is not quantified, and no autocorrelation, effective sample size, acceptance-rate curves, or sensitivity studies are provided. For MH chains on combinatorial spaces, these are essential.

4. The paper scales NFE by the number of drawn chain samples, but the practical cost per improved sample vs. baselines isn’t explored. Computation studies are not included, which can be important to show practical efficacy.

5. Some baselines are missing. Planning-augmented discrete diffusion and path-planning for masked diffusion methods are not compared. These explicitly do local search in discrete diffusion and represent trajectory-level guidance similar to MHDD’s local moves.

6. Comparison with training-based alignment for discrete diffusion method is missing. They are competitor paradigms for reward-guided generation, and adding side-by-side budget-matched comparisons can better justify the advantages.

7. Metrics commonly used other than reward, such as diversity, are missing, which are important to ensure MH steps don’t reduce diversity or exploit reward quirks. Current metrics are insufficient alone.

8. Ablations are very thin. No ablation studies are provided to show the contribution of key component. The method depends on designed sampling, short reverse length, chain length, etc., which should be studied.

**Questions:**

1. How sensitive is performance to narrower vs. wider time windows?

2. Do chains collapse to narrow modes?

3. Under equal wall-clock, how does MHDD compare to baselines?

---

> ### Author Response · Authors · 2025-11-25
> **Response to Reviewer a6b4, Part 1**
>
> We thank the reviewer for their detailed and insightful comments. We are pleased that the reviewer finds the cancellation of the intractable terms through the use of forward-backward diffusion proposals novel and that they find our method generalizable across discrete diffusion models.
>
> In this response, we respond to all the reviewer's comments as follows:
> - We address the reviewer's major concern about the computational of MHDD by additionally running and including results on MHDD **with batching** in Table 2 of the revised paper. We call this method **MHDD-B**, and show that this minor modification results in a significantly more efficient algorithm with minimal performance degradation. **(W4, Q3)**
> - We include additional comparisons with training-based methods, showing that MHDD outperforms these methods without any additional training **(W6)**, and include additional experiments on burn-in quantification **(W3)**, diversity **(W7, Q2)**, hyperparameter studies **(Q1)**, and ablation studies **(W8)** in the revision.
> - We clarify that the proof of the cancellation of intractable terms in the acceptance probability was already included in appendix B, and comment on the bias of the learned reverse model **(W1, W2)**. We also clarify that planning-augmented discrete diffusion is not used for reward-guided sampling as they do not incorporate reward signals during sampling **(W5)**.
>
> We thank the reviewer again for their comments and believe that our paper has improved by addressing and incorporating all their concerns and comments.

---

> ### Author Response · Authors · 2025-11-25
> **Response to Reviewer a6b4, Part 2**
>
> **W1. Analyses quantifying bias under approximate reverse kernels.** We thank the reviewer for their suggestion. Although it is true that using approximate reverse kernels may introduce bias into the computation of $p(x_0)$, we assume that the reverse kernels are sufficiently good approximations of the true reverse kernels. Exact quantification of the approximation error in diffusion models is still an active area of research, and such theoretical analysis is left for future work. Our results empirically suggest that the approximate reverse kernels are sufficient for reward-guided sampling.
>
> **W2. Where did $p_{pre}$ go in acceptance?** Please note that we have already included the explicit derivation of the cancellation of intractable terms in Appendix B, as stated in L353 in the main paper. For convenience, we have added a shortened proof in L340-L348 of the revised paper as well.
>
> **W3. Mixing/burn-in is not quantified** We thank the reviewer for raising this point. We have added plots of the autocorrelation function in Figure 5 in Appendix D.4 of the revision. As shown, the autocorrelation function quickly vanishes to zero within the first 2000 iterations and remains small from that point onwards. This indicates that the mixing time for MHDD is short. Although we burn the first half of the chain as a conservative estimate, these further results suggest that it may be possible to burn fewer samples.
>
> We also include acceptance rates for MDLM and USM below (and in Table 9 in Appendix D.4 of the revised paper):
>
> |       |       | QM9   |       |     |       | ZINC250K  |       |     | MPRA  |
> | ----- | ----- | ----- | ----- | --- | ----- | --------- | ----- | --- | ----- |
> |       | QED   | Rings | SA    |     | QED   | Rings     | SA    |     | HepG2 |
> | MDM   | 0.032 | 0.002 | 0.009 |     | 0.009 |     0.003 | 0.011 |     | 0.029 |
> | USM   | 0.510 | 0.330 | 0.346 |     | 0.725 |     0.597 | 0.711 |     | 0.038 |
>
> Although the acceptance rate for MDM is low due to MDM often generating invalid molecules, the acceptance rate is still acceptable, as we draw 128 samples from ~13k samples (<1%) after the burn-in period. On the other hand, USM has a much higher acceptance rate, which can be attributed to its higher validity rate due to its ability to correct errors from previous diffusion steps [1].
>
> **W4. Computation studies are not included.** We thank the reviewer for bringing to attention the wall-clock time performance of MHDD. Since MHDD is run sequentially, it is slow in terms of wall-clock time. In this revision, we propose a simple modification to MHDD by using batching to run multiple MHDD chains in parallel while decreasing the length of each chain to keep the total NFE unchanged. We refer to this as **MHDD-B** and include further results and description in Table 2 and L397 of the revised paper. Batching is a simple minor modification to the current algorithm which yields significant improvements in wall-clock time.
>
> We include the wall-clock time for generating 128 samples using USM guided with QED on the ZINC250K dataset in the table below. MHDD-B is significantly faster than MHDD, although the performance degrades slightly as shown in Table 2. Notably, MHDD-B yields the best performance, is faster than SMC and, at equal batch sizes, is roughly as fast as BoN as well. We also provide reward comparison with matched wall-clock times in our response to Q3, showing that MHDD-B outperforms all baselines on ring count and HepG2 rewards under equal wall-clock time.
>
> |            | MHDD  | MHDD-B | BoN   | BoN   | SMC   | SVDD  |
> | ---------- | ----- | ------ | ----- | ----- | ----- | ----- |
> | Time (s)   | 3029  | 334    | 138   | 316   | 359   | 264   |
> | Batch Size | 1     |     8  |   14  |   8   |   14  | 14    |
> | NFE/Sample | 1024  | 1024   | 1024  | 1024  | 1024  | 1024  |
> | Reward     | 0.916 | 0.917  | 0.910 | 0.913 | 0.766 | 0.719 |
>
> **W5. Planning-augmented discrete diffusion and path-planning for masked diffusion methods are not compared.** Planning-augmented discrete diffusion [2] and path-planning [3] are methods that improve the sampling of discrete diffusion models but are not related to reward-guided sampling since they do not incorporate any reward signals during sampling. Could the reviewer please clarify how these methods could be used for reward-guided sampling, and what they mean by "these explicitly do local search in discrete diffusion and represent trajectory-level guidance similar to MHDD’s local moves"? We are happy to engage in further discussions if the reviewer could clarify what they mean.

---

> ### Author Response · Authors · 2025-11-25
> **Response to Reviewer a6b4, Part 3**
>
> **W6. Comparison with training-based alignment for discrete diffusion method is missing.**
> We thank the reviewer for raising this point. We include additional results for training-based discrete diffusion CFG [1] in the table below. While it is unclear how to match the budget of training-based methods and our inference-time scaling method, we train the CFG model under the same settings as our pretrained models. As shown, MHDD outperforms CFG with significant improvements in ring count without requiring additional training. Furthermore, applying MHDD using the CFG model results in further improvements except in ring count for ZINC250K MDM. This is due to the MDM CFG model collapsing to narrow modes when trained on ZINC250K ring count, resulting in insufficient exploration or the space when applying MHDD. Indeed, one drawback of using training-based methods is deviation from the pretrained prior which may result in reward hacking. Our inference-time method, on the other hand, does not show mode collapse, as supported by the additional diversity metrics in Table 8 of appendix D.1 and as explained in our response to W7.
>
> |       |          | QM9   |         | ZINC250K |       |
> | ----- | -------- | ----- | ------- | -------- | ----- |
> |       |          | QED   | Ring    | QED      | Ring  |
> | MDM   | CFG      | 0.591 | 4.926   | 0.820    | 5.886 |
> |       | MHDD     | 0.610 | 10.000  | 0.910    | 8.091 |
> |       | CFG+MHDD | 0.631 | 10.608  | 0.927    | 7.214 |
> | USM   | CFG      | 0.607 | 4.843   | 0.903    | 5.613 |
> |       | MHDD     | 0.610 | 9.570   | 0.917    | 8.703 |
> |       | CFG+MHDD | 0.631 | 10.268  | 0.927    | 8.943 |
>
> **W7. Metrics commonly used other than reward, such as diversity, are missing.** We thank the reviewer for suggesting the additional metrics. We additionally report the diversity in Table 8 in the Appendix D.1 following the diversity metrics used in [4]. The diversity is computed by subtracting the mean pairwise similarity from 1. For molecule tasks, the pairwise similarity is computed using the Tanimoto similarity on the Morgan2 fingerprints. For MPRA, the pairwise similarity is computed as the cosine similarity of the one-hot encoding of the sequence.
>
> As shown in the table, MHDD does not significantly degrade the diversity in most cases. In all cases, the diversity remains sufficiently high. However, we also emphasize that the goal of reward-guided sampling is to produce high-reward samples, which will unavoidably lower the diversity when compared to the pretrained model. For ring count, the diversity is lower (**but still sufficiently high**) due to MHDD achieving significantly higher rewards than the other baselines.
>
> **W8. Ablations are very thin.** We thank the reviewers for raising the additional experiments. We include additional experiments on the hyperparameters $t_\text{lo}$ and $t_\text{hi}$ in Figure 4 in Appendix D.3 of the revised paper. We also include additional ablations on the number of reverse steps $M$ used in each MHDD iteration in the table below, and experiments on varying NFEs (chain length) in Figure 3 in Appendix D.3 of the revised paper. Our experiments indicate that the reward and diversity achieved by MHDD is not sensitive to the choice of $t_{lo}$ and $t_{hi}$. Please refer to our response to Q1 for more details on the sensitivity of our method to narrower or wider time windows.
>
> Varying $M$ does not significantly affect the rewards achieved by MHDD either, except when using large $M$ for the ring count reward. We hypothesize that this is due to limited compute since using $M=20$ for ZINC250K tasks results in only 88 iterations of MHDD, which may not be sufficient for harder tasks such as ring count. This hypothesis is also supported by the autocorrelation plot in Figure 5 in Appendix D.4 of the revised paper, which shows that the autocorrelation function for USM (green line) converges to zero more slowly in ring count compared to other rewards.
>
> Finally, MHDD consistently outperforms other baselines across various NFEs (Figure 3 in Appendix D.3) except for BoN for certain rewards and diffusion model. In particular, MHDD significantly outperforms all baselines in the ring count and HepG2 rewards.
>
> |     |       | QM9   |       |       | ZINC250K  |       |
> | --- | ----- | ----- | ----- | ----- | --------- | ----- |
> | M   | QED   | Ring  | SA    | QED   | Ring      | SA    |
> | 5   | 0.610 | 9.574 | 0.911 | 0.917 |     8.703 | 0.898 |
> | 10  | 0.601 | 9.392 | 0.913 | 0.910 |     8.477 | 0.901 |
> | 15  | 0.609 | 9.582 | 0.904 | 0.917 |     8.256 | 0.895 |
> | 20  | 0.601 | 9.062 | 0.919 | 0.917 |     6.885 | 0.897 |

---

> ### Author Response · Authors · 2025-11-25
> **Response to Reviewer a6b4, Part 4**
>
> **Q1. How sensitive is performance to narrower vs. wider time windows?** As shown in Figure 4 and Appendix D.3 of the revised paper, MHDD is not sensitive to the choice of $t_{lo}$ and $t_{hi}$. However, for ring count, the reward decreases as $t_\text{lo}$ increases. Intuitively, as $t_\text{lo}$ increases, each MHDD step changes the current $x_0$ sample significantly due to increased noise, resulting in greater exploration but less exploitation. For simpler rewards such as QED and SA, this does not significantly impact the average reward. However, for ring count where high-reward samples lie in extremely low-density regions of the pretrained model's learned distribution (refer to Figure 2 in the paper), it is necessary to choose smaller times to exploit the current $x_0$.
>
> **Q2. Do chains collapse to narrow modes?** We provide diversity metrics for samples taken from a *single* MHDD chain in the table below. We generate 128 samples for molecule tasks and 64 samples for MPRA. Details on the computation of the diversity metrics are described in L910-L915 in Appendix D.1 of the revised paper.
>
> As shown, the diversity for all rewards are sufficiently high, indicating that the chain has not collapsed to narrow modes. Although the diversity for the ring count reward is lower than the other metrics, this is due to molecules with high ring count lying in extremely low density regions of the pretrained distribution as shown in Figure 2 of the paper.
>
> |           |       | QM9   |       |       | ZINC250K  |       | MPRA  |
> | ----------- | ----- | ----- | ----- | ----- | --------- | ----- | ----- |
> |             | QED   | Ring  | SA    | QED   | Ring      | SA    | HepG2 |
> | Pretrained  | 0.923 | 0.923 | 0.923 | 0.880 |     0.880 | 0.880 | 0.748 |
> | BoN         | 0.896 | 0.891 | 0.914 | 0.863 |     0.869 | 0.842 | 0.749 |
> | SMC         | 0.925 | 0.923 | 0.928 | 0.873 |     0.876 | 0.874 | 0.748 |
> | SVDD        | 0.924 | 0.924 | 0.924 | 0.877 |     0.877 | 0.877 | 0.748 |
> | MHDD        | 0.882 | 0.828 | 0.872 | 0.861 |     0.843 | 0.844 | 0.675 |
>
> **Q3. Under equal wall-clock, how does MHDD compare to baselines?** In the table below, we include comparisons between MHDD-B and other baselines under equal wall-clock for ZINC250K and MPRA using USM. We ensure that MHDD-B uses less wall-clock time than other methods except for SVDD which is efficient at high NFEs but performs poorly. As shown, MHDD-B significantly outperforms all other baselines in ring count and HepG2 rewards while being comparable with BoN in QED and SA. When we match the batch sizes of MHDD-B and BoN, MHDD-B outperforms BoN in all rewards as well.
>
> |              |        | QED   |        |       |       |     |              |        | Ring  |       |       |       |
> | ------------ | ------ | ----- | ------ | ----- | ----- | --- | ------------ | ------ | ----- | ----- | ----- | ----- |
> |              | MHDD-B | BoN   | BoN    | SMC   | SVDD  |     |              | MHDD-B | BoN   | BoN   | SMC   | SVDD  |
> | Time (s)     | 334    | 334   | 359    | 359   | 264   |     | Time (s)     | 321    | 331   | 359   | 320   | 276   |
> | Batch Size   | 8      | 78    | 8      | 14    | 886   |     | Batch Size   | 8      | 78    | 8     | 14    | 886   |
> | NFE / Sample | 1024   | 5750  | 1024   | 1024  | 65536 |     | NFE / Sample | 1024   | 5750  | 1024  | 1024  | 65536 |
> | Reward       | 0.917  | 0.934 | 0.9133 | 0.766 | 0.719 |     | Reward       | 5.315  | 4.945 | 4.312 | 2.630 | 2.607 |
>
> |              |        | SA    |       |       |       |     |              |        | HepG2 |       |       |       |
> | ------------ | ------ | ----- | ----- | ----- | ----- | --- | ------------ | ------ | ----- | ----- | ----- | ----- |
> |              | MHDD-B | BoN   | BoN   | SMC   | SVDD  |     |              | MHDD-B | BoN   | BoN   | SMC   | SVDD  |
> | Time (s)     | 337    | 328   | 300   | 341   | 279   |     | Time (s)     | 74     | 80    | 103   | 398   | 246   |
> | Batch Size   | 8      | 78    | 8     | 14    | 886   |     | Batch Size   | 8      | 79    | 8     | 8     | 8     |
> | NFE / Sample | 1024   | 5750  | 1024  | 1024  | 65536 |     | NFE / Sample | 1000   | 10000 | 2000  | 1000  | 1000  |
> | Reward       | 0.912  | 0.919 | 0.892 | 0.792 | 0.755 |     | Reward       | 4.566  | 3.471 | 2.240 | 0.768 | 0.361 |
> ---
> [1] Schiff, Yair, et al. "Simple Guidance Mechanisms for Discrete Diffusion Models." The Thirteenth International Conference on Learning Representations.
>
> [2] Liu, Sulin, et al. "Think while You Generate: Discrete Diffusion with Planned Denoising." The Thirteenth International Conference on Learning Representations.
>
> [3] Peng, Fred Zhangzhi, et al. "Path planning for masked diffusion model sampling." arXiv preprint arXiv:2502.03540 (2025).
>
> [4] Li, Xiner, et al. "Derivative-free guidance in continuous and discrete diffusion models with soft value-based decoding." The Thirty-Ninth Annual Conference on Neural Information Processing Systems (2025).

---

### Author Response · Authors · 2025-11-25
**Simple Modification of MHDD to Support Batching**

**Based on the reviewers' shared concern over the computational efficiency of MHDD, we propose MHDD-B, a simple yet effective extension of MHDD by batching the generation process.** MHDD-B simply runs multiple MHDD chains in parallel while decreasing the length of each chain to keep the total NFE unchanged. This simple modification significantly reduces its wall-clock time without much degradation in performance. MHDD-B is faster than SMC and, at equal batch sizes, roughly as fast as BoN as well. **Additional results on MHDD-B are included in Table 2 of the revised paper, and details of MHDD-B are included in L397-L400 of the revised paper.**

To further analyze the computational efficiency of MHDD-B, we also include wall-clock time comparisons in **Table 3**, and comparisons of MHDD-B against the baselines under equal wall-clock time in **Tables 4 and 5** of the revision. These results show that
- MHDD-B speeds up MHDD by a factor of ~9x, with run times faster than SMC, and comparable to BoN when using the same batch size.
- MHDD-B significantly outperforms all other baselines in ring count and HepG2 rewards while being comparable with BoN in QED and SA. When we match the batch sizes of MHDD-B and BoN, MHDD-B outperforms BoN in all rewards as well.


**We hope that the inclusion of MHDD-B (Table 2) and our additional wall-clock time analysis (Tables 3,4,5) in the revised paper addresses the reviewers' concerns on the computational efficiency of our method.**

---

### Author Response · Authors · 2025-12-03

We thank the AC and reviewer's effort and time in evaluating our paper. We briefly summarize the rebuttals and discussion below.

## Reviewer a6b4
Reviewer a6b4 gave a negative score of 2, with the main concerns of computational efficiency of MHDD and the lack of convergence diagnostics and hyperparamater/ablation studies. During the rebuttals and discussion, we fully addressed the comments while introducing a slight modification of our algorithm during the discussion period that led to 9x speed up:
- **Computational Efficiency:** We propose MHDD-B, a simple yet effective extension of MHDD by batching the generation process. This simple modification results in a significantly more efficient algorithm with up to 9x improvement in wall-clock time. **Additional results on MHDD-B are included in Table 2 of the revised paper, and details of MHDD-B are included in L397-L400 of the revised paper.**
- **Lack of convergence diagnostics, hyperparameter/ablation Studies:** For convergence diagnostics, we added autocorrelation plots and report the acceptance rates in Figure 5 and Table 9 of Appendix D.4. We also include ablation/hyperparameter studies on the NFE (Figure 3, Appendix D.2) and time parameters (Figure 4, Appendix D.3) in the revised paper, showing that MHDD is not sensitive to these hyperparameters.

## Reviewer gv3v
Reviewer gv3v gave a positive score of 6, recognizing our contribution in "breaking the reliance of reward-guided sampling in discrete diffusion on intermediate rewards" by "combining the Metropolis-Hastings algorithm with the forward-backward processes of discrete diffusion to construct a Markov chain of clean samples." They raised concerns about the computation efficiency of our algorithm. We address their concern by proposing to batch MHDD running multiple chains in parallel and decreasing the length of each chain. We refer to this batched version of MHDD as **MHDD-B**. This simple modification significantly improves the computational efficiency with minimal degradation in performance. **Additional results on MHDD-B are included in Table 2 of the revised paper, and details of MHDD-B are included in L397-L400 of the revised paper.**

## Reviewer 1zUF
Reviewer 1zUF gave a score of 4, citing concerns about the convergence of MHDD, its efficiency, and the sensitivity of MHDD to its initialization. We fully addressed the comments while introducing a slight modification of our algorithm during the discussion period that led to 9x speed up:
- **Convergence:**
	- During the initial discussion, the reviewer had some confusion as to why using the Metropolis-Hastings algorithm guarantees convergence to the target distribution. We clarified that the Metropolis-Hastings (MH) algorithm has theoretically proven convergence guarantees (explained in Appendix A) and that MHDD inherits these convergence properties by using the MH algorithm.
	- The reviewer was satisfied with this response and followed up by asking for discussion on why MHDD can efficiently outperform BoN. We respond by elaborating that our design choice involves sampling $t_0 \sim [t_\text{lo}, t_\text{hi}]$ which allows for a controllable exploration-exploitation tradeoff: Larger $t_0$ leads to increased exploration, which helps convergence by preventing the process from being stuck in local minima; smaller $t_0$ results in more local changes which exploits information on the current $x_0$ which gives advantage over BoN.
- **Efficiency:** We address concerns about the efficiency of MHDD by plotting reward against NFE and showing that MHDD consistently outperforms other baselines. **We also additionally modify MHDD to support batching, resulting in MHDD-B.** This simple modification significantly reduces the wall-clock time of MHDD with only marginal degradation in performance.
- **Sensitivity to Initialization:** We clarify that the samples generated in each experiment are taken from multiple Markov chains, each with a different initial sample. We also added 95% confidence to Table 2 as the reviewer suggested. Finally, when using MHDD-B, different initializations are used for each chain in the batch, further mitigating sensitivity to the initialization.

## Reviewer 7t7b
Reviewer 7t7b gave a negative score of 2, with the major concern being the theoretical soundness of MHDD. They point out that the proof of the simplified acceptance probability assumed a fixed $x_t$ but in practice, $x_t$ is resampled at each step. We fully address their concern by including additional explanation and derivation showing that the Markov chain induced by fixing $x_t$ is equivalent to the Markov chain induced by sampling $x_t$ at each step. This explanation and derivation are included in L307-319 of the revised paper.

---

### Meta-Review · Area_Chair_b1XY · 2025-12-22

**Summary:**

The paper introduces MHDD, a method that guides sampling using only clean rewards with a Markov chain of clean samples that converges to a target reward-weighted distribution. To make the Metropolis-Hastings acceptance ratio tractable, the authors corrupt the current clean sample $x_0$ to a random noise level $t$, and run the diffusion reverse process from $x_t$ to generate a proposal $x'_0$. This allows the sampler to perform local search on the clean manifold, utilizing accurate reward signals rather than noisy approximations.

During the rebuttal, reviewers challenged the method's slow, sequential nature. The authors responded by introducing MHDD-B, which runs multiple shorter chains in parallel instead of one long sequential chain. MHDD-B was shown to be faster than SMC and comparable in speed to Best-of-N (BoN), effectively removing the primary efficiency bottleneck while retaining most of the original performance.

However, while the paper frames the method as having exact convergence guarantees, the derivation of the acceptance probability relies on the assumption that the learned reverse diffusion model is perfect. In practice, the neural network approximation introduces a bias that technically breaks the exact detailed balance. This limitation was acknowledged but not quantified by the authors during the review.

Moreover, as reviewers pointed out, the method’s success likely comes from the proposal distribution exploring the neighborhood of good samples (a “warm start” effect) rather than the rigorous rejection sampling of MH. To isolate the benefits of the MH step, comparisons against a baseline of parallelized BoN with naive perturbations on the best clean samples would be valuable. While the authors provided a hypothesis on why and when MHDD outperforms BoN, this mechanism warrants further exploration and confirmation.

In conclusion, the AC acknowledges the empirical strength of the paper but has concerns regarding the theoretical rigor of its connection to MH and the technical depth regarding the source of its benefits. Therefore, the AC recommends rejecting the paper to allow the authors to improve the theoretical framing and analysis for resubmission to a future venue.

**Reviewer Concerns:**

**Addressed Concerns**
- Computational Efficiency & Wall-Clock Time: Concerns raised by Reviewers a6b4, gv3v, and 1zUF regarding the slow, sequential nature of the algorithm were effectively addressed. The authors introduced MHDD-B (Batched MHDD), which runs parallel chains to match the total NFE budget. This modification yielded speedup, demonstrating performance faster than SMC and comparable to Best-of-N (BoN) in wall-clock time.
- Lack of Diagnostics & Ablations: Reviewers a6b4 and 1zUF requested standard MCMC diagnostics. The authors successfully added autocorrelation plots showing fast mixing (within 2000 iterations) 2and sensitivity analyses demonstrating robustness to hyperparameters $t_{lo}$ and $t_{hi}$.
- Comparison with Baselines: Requests for comparisons against training-based methods (Reviewer a6b4) and questions regarding why intermediate-reward baselines like SMC fail (Reviewer 1zUF) were addressed with new experiments and detailed explanations regarding the noise in intermediate rewards.

**Outstanding Concerns**
- Theoretical Validity of the Acceptance Ratio: Reviewers 7t7b and a6b4 identified a mathematical issue, where the acceptance ratio is derived by fixing the intermediate noise state $x_t$ rather than integrating over it (as required for the marginal proposal distribution). The authors' defense—that the generation process is time-homogeneous—does not mathematically justify using a single-path ratio for the acceptance step without assuming a perfect model.
- Source of Empirical Gains: Reviewer 1zUF suggested that the method's success likely stems from a heuristic "warm start" effect (local exploration around good samples) rather than the rigorous application of Metropolis-Hastings rejection sampling. While the author gave some reasoning, these need to be formally incorporated in the paper and show empirical studies.

**Reviewer Scores:**

- Reviewer gv3v (6 to 8): This reviewer was already positive, citing the clean reward idea as a strength. Their primary reservation was efficiency. The introduction of MHDD-B directly resolved this blocker, likely moving them to an accept.
- Reviewer 1zUF (4 to 4): This reviewer was borderline but engaged. They acknowledged the “warm start” explanation as plausible and appreciated the efficiency updates. However, their fundamental concerns regarding convergence would likely keep their score.
- Reviewer a6b4 (2 to 4): While the authors addressed this reviewer's empirical requests, the reviewer explicitly penalized the paper for Soundness due to the bias in the approximate reverse kernel. Since the theoretical aspect remains unquantified, this reviewer would likely improve their score to 4 maximum.
- Reviewer 7t7b (2 to 2): This reviewer’s objection was rigorously mathematical. The authors’ rebuttal defended the formulation rather than correcting it or characterizing it as an approximation. The score would likely remain unchanged.

---

### Decision · Program_Chairs · 2026-01-26

Reject